# Elucidating target specificity of the taccalonolide covalent microtubule stabilizers employing a combinatorial chemical approach

Lin Du [1,2,6]*, Samantha S. Yee [3,6], Karthik Ramachandran [4] & April L. Risinger [3,5]*

The taccalonolide microtubule stabilizers covalently bind β-tubulin and overcome clinically relevant taxane resistance mechanisms. Evaluations of the target specificity and detailed drug–target interactions of taccalonolides, however, have been limited in part by their irreversible target engagement. In this study, we report the synthesis of fluorogenic taccalonolide probes that maintain the native biological properties of the potent taccalonolide, AJ. These carefully optimized, cell-permeable probes outperform commercial taxane-based probes and enable direct visualization of taccalonolides in both live and fixed cells with dramatic microtubule colocalization. The specificity of taccalonolide binding to β-tubulin is demonstrated by immunoblotting, which allows for determination of the relative contribution of key tubulin residues and taccalonolide moieties for drug–target interactions by activity-based protein profiling utilizing site-directed mutagenesis and computational modeling. This combinatorial approach provides a generally applicable strategy for investigating the binding specificity and molecular interactions of covalent binding drugs in a cellular environment.

[1] Department of Chemistry and Biochemistry, The University of Oklahoma, Norman, OK, USA. [2] Institute for Natural Products Applications and Research Technologies, The University of Oklahoma, Norman, OK, USA. [3] Department of Pharmacology, The University of Texas Health Science Center, San Antonio, TX, USA. [4] Department of Medicine, Division of Nephrology, The University of Texas Health Science Center, San Antonio, TX, USA. [5] Mays Cancer Center, The University of Texas Health Science Center, San Antonio, TX, USA. [6] These authors contributed equally: Lin Du, Samantha S. Yee. *email: Lin.Du-1@ou.edu; risingera@uthscsa.edu

Many successful therapeutics, including aspirin, β-lactam antibiotics, esomeprazole (Nexium), and clopidogrel (Plavix)[1,2] bind covalently to their drug targets. However, the irreversible nature of their binding prompts safety concerns due to potential off-target reactivity and unanticipated side effects. Therefore, one of the most critical steps in the covalent drug discovery process is the effective evaluation of their target specificity and assessment of useful derisking strategies[3,4]. The development of modern 'targeted covalent inhibitors' (TCIs)[5,6] has led to significant progress including the successful launch of several preclinical and clinical studies for covalent EGFR inhibitors, such as the FDA approved afatinib (Giltrif) and osimertinib (Tagrisso), which exhibited promising therapeutic effects against resistant cancer models expressing EGFR mutations[7–9]. The systematic studies of TCIs have also revealed that the safety of covalent drugs needs to be evaluated on a case-by-case basis and the complexity of the covalent systems often urge innovative approaches[10–12] as necessary complements to conventional preclinical and clinical studies.

The taxane class of microtubule stabilizers is a mainstay in the clinical treatment of solid tumors even in the era of targeted therapy and immunotherapy[13–16]. However, a major limitation of the taxanes is acquired drug resistance. The taccalonolides are a class of microtubule stabilizers that covalently bind β-tubulin[17,18] and effectively circumvent clinically relevant models of resistance to taxanes both in vitro and in vivo[19–21]. Despite their promising therapeutic potential, the covalent nature of taccalonolide binding has hampered our ability to perform detailed binding studies using conventional approaches[22,23]. Consequently, it urged us to develop a functional and rigorous activity-based approach to elucidate the target specificity and drug–target interactions of the taccalonolides. Here we describe the synthesis and optimization of a fluorogenic taccalonolide probe, Flu-tacca-7 (11). This stable, cell-permeable probe is used for activity-based protein profiling (ABPP) in human cancer cell lines to confirm the specificity of covalent binding of the taccalonolides to β-tubulin and evaluate key β-tubulin residues and taccalonolide moieties that mediate taccalonolide-tubulin binding. Flu-tacca-7 (11) represents a class of irreversible microtubule labeling probes that are superior to commercially available options, providing a valuable tool for cellular evaluations of this important drug target.

## Results

**Optimization of fluorogenic taccalonolide-based probes.** While we were exploring strategies for generating an optimal taccalonolide-based chemical probe, Wang *et al.* reported the crystal structure (PDB ID: 5EZY) of tubulin complexed with taccalonolide AJ (2) (Fig. 1a, c)[18]. The authors proposed an unusual reaction mechanism intending to explain the covalent bond formation between the 22,23-epoxy moiety of 2 and β-tubulin D226. However, on account of the existing conflicts in the literature about the absolute configuration of the 22,23-epoxy group[21,24], we unambiguously verified the 22 R,23 R configuration of 2 by single-crystal X-ray diffraction analysis (Fig. 1b)[25]. Therefore, the opening of the 22,23-epoxy group of 2 is likely facilitated via direct nucleophilic attack by the carboxylate of β-tubulin D226 (Fig. 1d)[26]. This epoxide opening mechanism was supported by covalent docking of 2 into β-tubulin using CovDock affording a lowest-energy docking model that perfectly matched the 5EZY crystal structure (RMSD = 0.221, Fig. 1c)[27]. Analysis of the docking data also disclosed that several other key β-tubulin residues (e.g. K19, H229, R278, L217, L219, and T223) were likely to play important roles in mediating the binding affinity of 2 (Fig. 1e). But more importantly, the careful examination of the chemical environment in the binding pocket revealed that the C-6 ketone group of 2 was positioned relatively remote from all β-tubulin residues and was not involved in any inter- and intra-molecular interactions (Fig. 1e). Thus, the taccalonolide C-6 position was identified as an optimal site for linker/payload

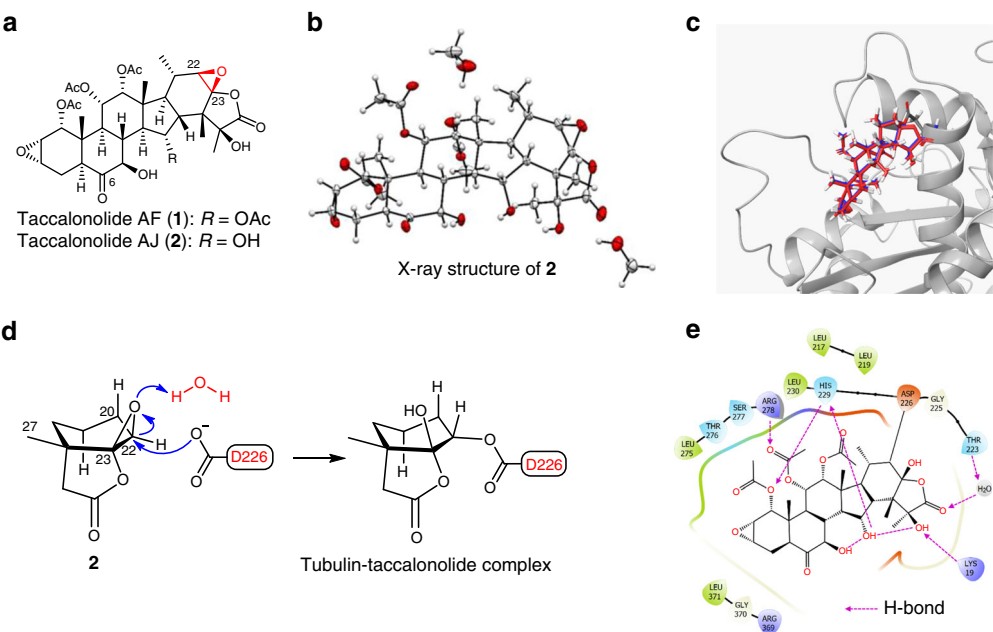

**Fig. 1 The taccalonolide microtubule stabilizers covalently bind β-tubulin. a** Structures of taccalonolides AF (**1**) and AJ (**2**) showing verified absolute configuration of the 22,23-epoxy moiety. **b** The ORTEP drawing of the single-crystal X-ray structure of **2** (CCDC ID: 1907790). **c** The published crystal structure (PDB ID: 5EZY) of **2** (red) bound to β-tubulin was superimposed with a model of **2** (blue) docked into β-tubulin (RMSD = 0.221) generated by CovDock. **d** The proposed reaction mechanism between **2** and β-tubulin D226. **e** The key tubulin residues that mediate the binding affinity of **2** in the docking model structure of **c**. The residues within a radius of 3.5 Å from **2** are displayed. The key H$_2$O molecule bridging **2** and T223 was retained as it improved the accuracy of docking experiments.

**Fig. 2 Structures of the semi-synthetic taccalonolide-based fluorescent probes 3–12.** The taccalonolide probes synthesized and evaluated in this study are shown. Synthetic schemes and methods can be found in Supplementary Figs. 82–96 and the Supplementary Methods, respectively.

conjugation to generate a stable taccalonolide probe that was likely to maintain the native biological properties of **2**.

Our strategy to functionally characterize taccalonolide-tubulin binding using fluorescent taccalonolide probes is based on the well-established activity-based protein profiling (ABPP) approach[11], which facilitates determination of drug–target interactions in a cellular context and is particularly suited to compounds that covalently bind their targets. Initial attempts to generate a stable taccalonolide probe by modification of taccalonolide C-6 led to the synthesis of Flu-tacca-1 (**3**) (Fig. 2), a fluorescent probe that enabled direct visualization of the taccalonolides in live cancer cells[28]. However, there were several disadvantages of this probe, including the lability of the ester-based linker, the weak micromolar cellular potency, poor fluorescence properties due to the masked phenolic hydroxyl group of the fluorescein moiety, and high background fluorescence that necessitated removal of excess probe from the media prior to imaging. These limitations urged us to generate additional taccalonolide probes that were more suitable for ABPP studies. By means of a survey of various strategies to effectively modify the C-6 position of the taccalonolides, we identified a

convenient approach to convert taccalonolide B (**13**) to its C-6 amino analogue **14** through reductive amination (Fig. 3). The employment of the 4 Å molecular sieve as a dewatering agent in the reaction played a vital role in suppressing the formation of the C-6 hydroxy side product (<5% yield)[28]. With **14** in hand as a key intermediate, we were able to generate a set of stable amide-based fluorescent/fluorogenic probes **4**–**12** employing varying linker length, fluorescent moieties, and prodrug strategies (Fig. 2). The optimization of the taccalonolide probes was guided by evaluation of their biological properties and comparison with the untagged taccalonolide AJ (**2**) in a series of cellular and biochemical experiments (Table 1, Figs. 4, 5).

The synthesis of the amide-based probes, Flu-tacca-2 (**4**) and Flu-tacca-3 (**5**), was inspired by the structure of the commercial taxane-based probe, Tubulin Tracker Green (Thermo Fisher Scientific, Oregon Green™ 488 Taxol, Bis-Acetate). Specifically, a protected fluorescent moiety (Oregon Green 488 for **4** and fluorescein for **5**, diacetyl form) was conjugated with **14** via a β-alanine linker. According to the manufacturer's description[29], the diacetyl protection on Oregon Green is intended to quench fluorescence prior to intracellular hydrolysis of the acetyl groups

**Fig. 3 Synthesis of Flu-tacca-7 (11).** The synthetic scheme for the generation of the most potent dipivaloyl-protected taccalonolide fluorescent probe. Detailed methods can be found in Supplementary Methods.

**Table 1 Concentrations (nM) of taccalonolide probes that cause a 50% decrease in the proliferation (GI$_{50}$) of HeLa or SK-OV-3 cells.**

| Compound | HeLa | SK-OV-3 |
|---|---|---|
| 2 | 8.5 ± 0.1 | 6.2 ± 1.4 |
| 3 | 2,100 ± 200[26] | ND |
| 4 | >20,000 | >20,000 |
| 5 | 12,200 ± 1,700 | 11,800 ± 2,100 |
| 6 | 8,300 ± 600 | 10,800 ± 1,600 |
| 7 | >20,000 | >20,000 |
| 8 | 1,500 ± 400 | 4,500 ± 900[a] |
| 9 | 740 ± 60 | 970 ± 60 |
| 10 | >20,000 | >20,000 |
| 11 | 31 ± 2 | 47 ± 8[a] |
| 12 | 3,000 ± 200[a] | 7,200 ± 400 |

[a]data from 4 independent experiments.
GI$_{50}$ values were obtained from three independent experiments (unless otherwise noted) each performed in triplicate and presented as mean ± SEM. Source data are provided as a Source Data file.

by esterases to decrease background fluorescence of any unincorporated probe. However, the Oregon Green probe **4** was rapidly hydrolyzed even in methanol solutions likely due to the relatively low p$K_a$ of the Oregon Green moiety (Oregon Green, p$K_a$ 4.8 and fluorescein, p$K_a$ 6.5, unprotected form)[30] and did not have antiproliferative potency up to a concentration of 20 μM. In contrast, the fluorescein probe **5** showed improved stability in organic solutions but was readily hydrolyzed to yield the deprotected form **6** in a 50% methanol/PBS solution and in the RPMI 1640 medium (Supplementary Fig. 1a–c, g, and h). Both **5** and **6** were over 125-fold less potent than the untagged **2** against HeLa cervical, SK-OV-3 ovarian, and two triple-negative breast (HCC1806 and HCC1937) human cancer cell lines as determined by the concentration that caused a 50% inhibition of proliferation as compared to vehicle treated controls (GI$_{50}$) (Table 1 and Supplementary Fig. 2). Furthermore, both **5** and **6** showed strong extracellular fluorescence in the cell culture medium that remained at a low level even after excess probe was removed from the medium prior to imaging (Fig. 4a). Together these results indicated that the diacetyl protection of fluorescein or

Oregon Green is not an optimal strategy for generating stable, cell-permeable taccalonolide probes for imaging applications.

As we were exploring efficient separation protocols for the synthetic isomeric mixture 5(6)-carboxyfluorescein (**15**) (Fig. 3), we noticed that the dipivaloyl protection of **15** to give **16** resulted in a complete quenching of carboxyfluorescein fluorescence in aqueous solutions and enabled baseline-separation of the two isomers by HPLC using a preparative C18 column. Intrigued by the quenched fluorescence of 5-carboxyfluorescein dipivalate (**16**), we synthesized the taccalonolide probe Flu-tacca-5 (**8**), a dipivaloyl-protected analogue of **5** and **6** (Fig. 2). As expected, compound **8** exhibited exceptional hydrolytic stability in both organic and aqueous solutions as well as in the cell culture medium (Supplementary Fig. 1). Moreover, the dipivaloyl protection effectively quenched the fluorescence of the extracellular probe in cell culture media enabling comparable intracellular staining in live cells before and after excess probe was removed from the medium with virtually no background fluorescence (Fig. 4a). Further intracellular localization studies indicated **8** promoted cellular microtubule stabilization and colocalized with β-tubulin immunofluorescence in both fixed HCC1937 and HeLa cell lines (Fig. 4b). Despite the improved visualization of the dipivaloyl probe in both live and fixed cells, the micromolar antiproliferative potency of **8** (HeLa, GI$_{50}$: 1.5 μM; SK-OV-3, GI$_{50}$: 4.5 μM) as compared to the nanomolar potency of untagged **2** (HeLa, GI$_{50}$: 8.5 nM; SK-OV-3, GI$_{50}$: 6.2 nM) prompted us to further optimize the intracellular potency of dipivaloyl-protected probes.

Inspired by a recent publication showing improved antiproliferative activities can be achieved for taxane-based probes by replacing a β-alanine linker with a shorter glycine linker[31], we synthesized two dipivaloyl-protected taccalonolide probes, including Flu-tacca-6 (**9**) utilizing a glycine linker and Flutacca-7 (**11**) featuring direct conjugation of the fluorescein moiety with the taccalonalide skeleton by an amide bond (Figs. 2, 3). Indeed, a progressive shortening of the linker between the taccalonolide and the dipivaloyl-protected fluorescein moiety enhanced the antiproliferative potency of **9** and **11** as compared to **8** against HeLa and SK-OV-3 cell lines (Table 1). In particular, the direct drug-fluorophore conjugation in **11** led to GI$_{50}$ values of 30–50 nM, a 50-fold improvement in potency as compared to **8** and <10-fold difference as compared to the untagged **2** (Table 1).

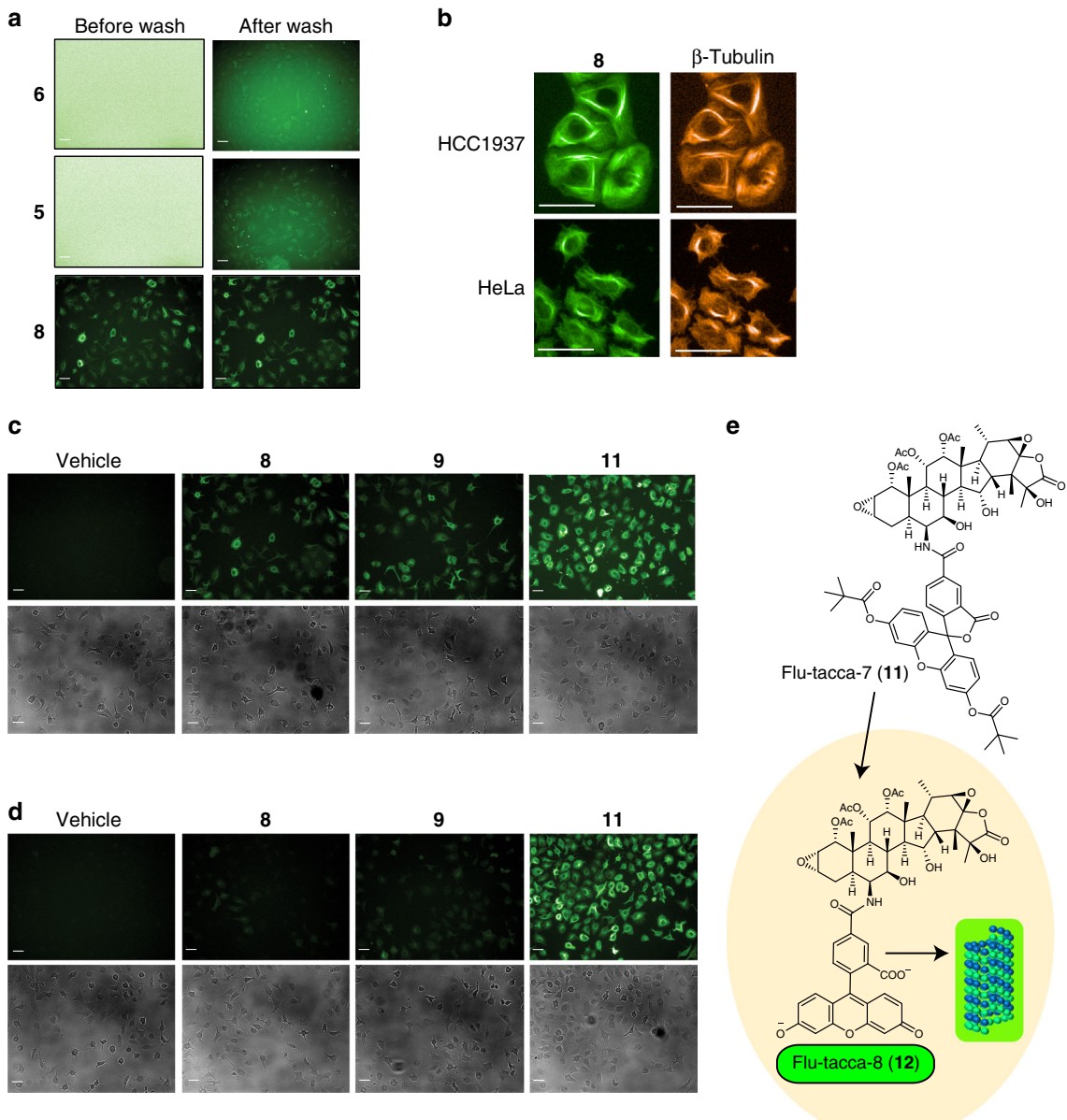

**Fig. 4 Optimization of taccalonolide-fluorescein probes. a** Visualization of unprotected (**6**), acetyl-protected (**5**), and pivaloyl-protected (**8**) probes in live SK-OV-3 cells 5 h after the addition of 5 μM probe either before (left) or after (right) medium containing probe was removed and replaced by fresh medium. **b** Co-localization of **8** (green) with β-tubulin immunofluorescence (orange) in fixed HCC1937 and HeLa cells after 10 μM probe addition for 6 h. **c**, **d** Pivaloyl-protected taccalonolide-fluorescein probes with decreasing linker sizes (left to right) were added to SK-OV-3 human ovarian cancer cells at a concentration of 5 μM **c** or 0.5 μM **d** and incubated for 5 h prior to imaging. The same image acquisition and processing conditions were used for each image to directly compare relative fluorescence properties (top rows) along with brightfield images of the field (bottom row). **e** A no-wash fluorogenic labeling system for cellular tubulin employing the dipivaloyl-protected taccalonolide probe Flu-tacca-7 (**11**), which is cleaved to generate Flu-tacca-8 upon cellular entry where it can bind cellular microtubules. Scale bars = 50 μm for all images. Source data are provided as a Source Data file.

Furthermore, the potent probe **11** had significantly improved intracellular fluorescence brightness as compared to **8** and **9** when used at equimolar concentrations and imaged under identical acquisition and visualization conditions (Fig. 4c, d and Supplementary Fig. 3). Together, our data demonstrate that Flu-tacca-7 (**11**) represents a cell-permeable, fluorogenic probe that combines the potent antiproliferative activities of taccalonolide AJ with excellent fluorescence properties, including the complete quenching of fluorescence until the dipivaloyl moieties are cleaved, likely by intracellular esterases (Fig. 4e), for the imaging of tubulin in both live and fixed cells.

To determine whether the differences in cellular potency among the taccalonolide probes correlated with target engagement, we evaluated the potency and efficacy of the unprotected probes **6** and **12** and dipivaloyl-protected probes **8** and **11** as compared to **2** in a biochemical tubulin polymerization assay. The untagged taccalonolide AJ (**2**) promoted the polymerization of purified tubulin (20 μM) in a concentration-dependent manner over a range of 5–20 μM (Fig. 5a), as previously described[17]. Unexpectedly, the most potent taccalonolide probe in cellular assays, **11**, only slightly enhanced microtubule polymerization at the highest tested concentration (20 μM) (Fig. 5a). In contrast, **12**,

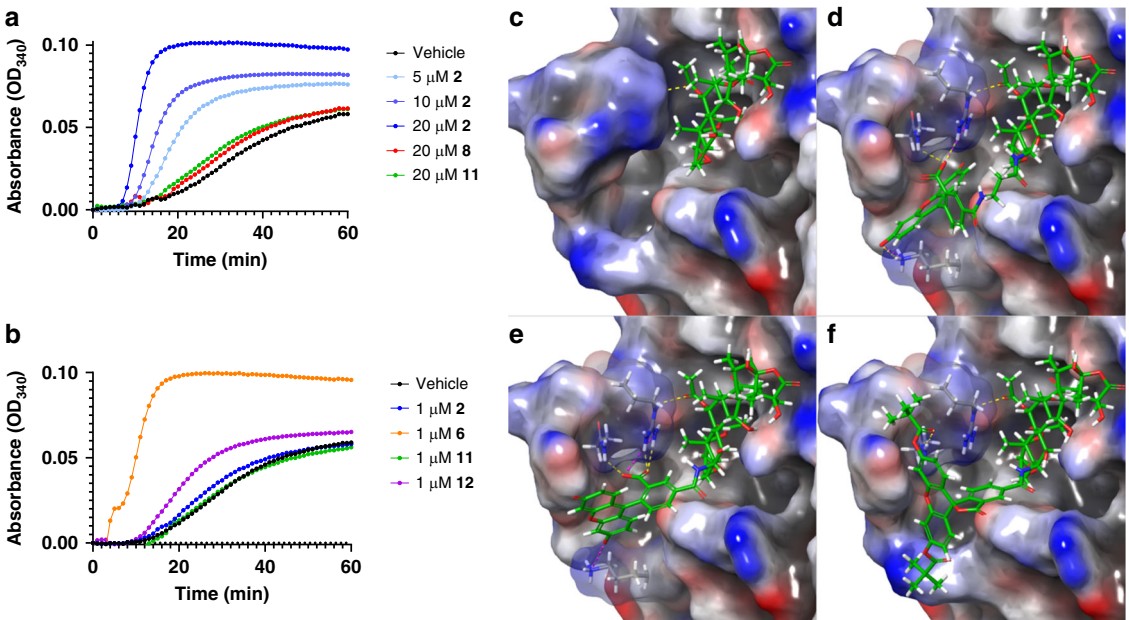

**Fig. 5 Taccalonolide probes engage additional β-tubulin contacts. a, b** Effects of free taccalonolide AJ (**2**) and taccalonolide-fluorescein probes (**6**, **8**, **11**, **12**) on the polymerization of purified porcine tubulin. **a** Concentration-dependent (5–20 μM) effects of **2** on tubulin polymerization with vehicle in black and **2** in shades of blue from lightest to darkest as concentration increases. The introduction of pivaloyl protecting groups in **8** (red) and **11** (green) diminish this activity even at equimolar concentrations with tubulin (20 μM). **b** The deprotected taccalonolide probes **6** (orange) or **12** (purple) are more potent than **2** (blue) or **11** (green) in their ability to polymerize purified tubulin. **c–f** Comparison of the published structure 5EZY (**2** bound to β-tubulin) **c** with a model of β-tubulin docked to taccalonolide probes (**6**) **d**, (**12**) **e**, and (**11**) **f**. The selected H-bonds and salt bridges are displayed as yellow and magenta dashed lines, respectively. Source data are provided as a Source Data file.

the deprotected analogue of **11**, was more potent than **11** or **2** in this biochemical assay (Fig. 5b). A more dramatic case was observed for the dipivaloyl-protected **8**, which did not promote polymerization even at 20 μM, as compared to its deprotected analogue **6**, which promoted robust polymerization at 1 μM that was equivalent to the effect of 20 μM of **2** (Fig. 5a, b). Due to the covalent nature of the interaction of the taccalonolide probes with tubulin, we were able to determine that the time course of binding of **12** to purified tubulin correlated with microtubule polymerization at both 1 and 20 μM (Supplementary Fig. 4). This confirms that the lag time associated with tubulin polymerization seen with the taccalonolides (which is distinct from the immediate polymerization observed with the taxanes) is associated with slow initial binding of the drug that then increases rapidly concomitant with microtubule nucleation[32].

Covalent docking using CovDock was employed to rationalize the increased potency of the taccalonolide probes **6** and **12** for biochemical tubulin polymerization as compared to the unmodified taccalonolide **2**. The top 10 low-energy poses (ligand-receptor binding models) were generated for **6**, **8**, **11** and **12**, respectively, that were docked into the optimized 5EZY structure. The representative lowest-energy pose obtained from each docking experiment was displayed (Fig. 5c–f). The taccalonolide core structure of **12** (Fig. 5e) was correctly positioned into the taccalonolide binding pocket (based on the 5EZY crystal structure of **2**, Fig. 5c) for each of the top 10 low-energy β-tubulin binding models. Interestingly, the fluorescein moiety of **12** occupied an adjacent binding pocket on β-tubulin close to the M-loop affording additional interactions with β-tubulin residues via hydrophobic interactions, H-bonds, and/or salt bridges (Fig. 5e). A similar binding mode was predicted for **6** (Fig. 5d), suggesting that the enhanced ability of **6** and **12** to promote microtubule stabilization in biochemical assays could be attributed to improved binding affinity to β-tubulin afforded by these additional contacts. In contrast, the lowest-energy β-tubulin

binding model for **11** clearly showed that the two dipivaloyl protecting groups hampered the ability of the fluorescein moiety to be positioned into this additional binding pocket and the taccalonolide core structure was only correctly positioned in 2 of the 10 low-energy poses (Fig. 5f). In the other 8 low-energy poses for **11**, the taccalonolide core structures were "forced" into the binding pocket but the structures were significantly rotated implying those binding models were not reliable and suggesting that **11** was poorly bound to β-tubulin. Similarly, the taccalonolide core structure of **8** failed to correctly fit into the taccalonolide binding pocket in any of the 10 low-energy β-tubulin binding models consistent with the inability of **8** to enhance tubulin polymerization in biochemical assays as compared to vehicle controls (Fig. 5a). Thus, the analysis of taccalonolide probe binding based on covalent docking data was qualitatively consistent with the biochemical tubulin polymerization assay. Although more comprehensive computational and experimental analyses would be required to provide a more complete understanding of the mode of action for the taccalonolide probes, the current binding analysis provides guidance for future efforts to generate taccalonolide analogues with improved binding affinity to tubulin by engaging an additional binding pocket adjacent to the M-loop.

The finding that the dipivaloyl-protected fluorogenic probes **11** and **8** are unable to directly interact with and polymerize tubulin or exhibit fluorescence in the medium, but effectively stabilize microtubules and exhibit fluorescence in cellular assays suggests that upon cellular entry these probes are activated via the hydrolysis of the dipivaloyl groups, likely by cellular esterases, (Fig. 4e). Indeed, when HCC1937 cells were treated with the dipivaloyl protected 22,23-ene analogue (**10**) of Flu-tacca-7 (**11**) to prevent irreversible binding to its target, its deprotected, fluorescent form **28** was able to be detected from cellular lysates by LCMS analysis (Supplementary Fig. 5). Thus, the dipivaloyl protective groups serve two distinct roles for the taccalonolide

probes: effectively quenching extracellular fluorescence and retaining the taccalonolides in a binding-deficient form prior to entry into cells where fluorescence and tubulin binding can both occur.

**Target specificity of the taccalonolides.** The generation of the potent fluorogenic taccalonolide probe Flu-tacca-7 (**11**) provides the opportunity to evaluate the covalent binding specificity of the taccalonolides in cell-based assays and fully define the key structural elements of both taccalonolides and β-tubulin that are essential for drug binding. The 22,23-epoxide moiety has been suggested to be critical for the antiproliferative and microtubule stabilizing effects of the taccalonolides[18,24,33]. Thus, we compared the antiproliferative activities, cellular localization, and proteome reactivity profiles of Flu-tacca-7 (**11**) and its 22,23-ene analogue **10**. The single replacement of 22,23-epoxide in **11** by 22,23-ene in **10** completely abrogated the antiproliferative activities in all four human cancer cell lines (i.e. HeLa, SK-OV-3, HCC1806, and HCC1937) up to 20 μM (Table 1 and Supplementary Fig. 2). Additionally, **11** promoted a concentration-dependent (0.05–5 μM) increase in the polymerization of cellular tubulin, as detected by immunofluorescence and confocal imaging (Fig. 6a, red), in line with its antiproliferative potency in HCC1937 cells (Supplementary Fig. 2). The intrinsic fluorescence of **11** (Fig. 6a, green) colocalized completely with the cellular microtubules. In contrast, the 22,23-ene analogue **10** failed to either promote cellular tubulin polymerization (Fig. 6a, red) or colocalize with microtubules (Fig. 6a, green) at 5 μM. This 22,23-epoxide-dependent cellular microtubule stabilization and colocalization was also observed by comparing the localization of 22,23-epoxide and 22,23-ene probes in both live and fixed cells (Supplementary Fig. 6a, b). On account of the covalent nature of the taccalonolide interaction with tubulin[17], the protein binding profiles of **10** and **11** were also evaluated by immunoblotting under denaturing and reducing conditions that would disrupt non-covalent interactions using an anti-fluorescein antibody. While the cells treated with **10** or **11** showed equivalent expression of β-tubulin (Fig. 6b), a single 50 kDa fluorescein-containing band was readily identified in cells treated with **11**, but not **10** (Fig. 6c). The results demonstrated that the 22,23-epoxy group was essential for the taccalonolide probes to efficiently form a covalent bond with its major cellular target of β-tubulin in the cellular environment, demonstrating the specificity of this interaction. Similar results were obtained from the comparison of Flu-tacca-5 (**8**) and its 22,23-ene analogue **7** (Supplementary Fig. 6c, d). Together, these data strongly demonstrate that the 22,23-epoxy moiety is critical for the covalent reaction of the taccalonolides with β-tubulin.

Inspired by the ability to detect the cellular binding of Flu-tacca-7 (**11**) to endogenous β-tubulin by immunoblot, we engineered a system to evaluate the relative contribution of individual β-tubulin amino acid residues to taccalonolide binding by performing site-directed mutagenesis on an ectopically expressed β-tubulin construct tagged with GFP at the C-terminus to distinguish it by size on an immunoblot. In order to test the feasibility of our approach, we first mutagenized the D226 residue of β-tubulin to either an asparagine or alanine (Supplementary Table 9). HeLa

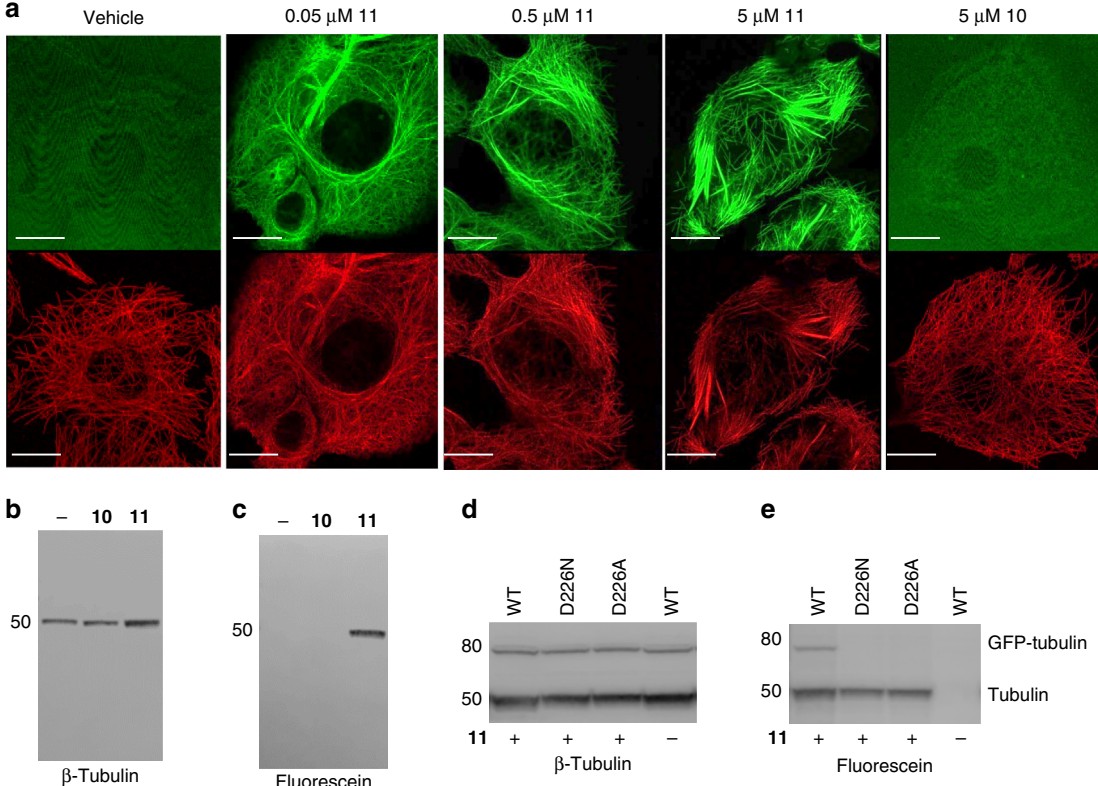

**Fig. 6 The taccalonolide epoxide and β-tubulin D226 are critical for covalent binding. a** HCC1937 cells were treated with 0.05–5 μM taccalonolide probes with (**11**) or without (**10**) the 22,23-epoxide for 24 h. Colocalization of taccalonolide probes (green) with β-tubulin immunofluorescence (red) was evaluated by confocal imaging. **b, c** HCC1937 cells treated with 5 μM **10** or **11** for 6 h were lysed and subjected to immunoblotting using an anti-β-tubulin antibody **b** or an anti-fluorescein antibody **c**. **d, e** HeLa cells were transfected with GFP-tagged TUBB1 constructs with indicated mutations then treated with 1 μM **11** for 8 h. Cell lysates were harvested for immunoblotting. β-tubulin immunoblotting **d** showed the expression of endogenous tubulin (50 kDa) and the GFP-tubulin constructs (77 kDa). **e** An anti-fluorescein antibody was used to detect **11** bound to endogenously expressed β-tubulin (lower band, 50 kDa) and the GFP-tubulin constructs (upper band, 77 kDa). Scale bars = 10 μm for all images. Source data are provided as a Source Data file.

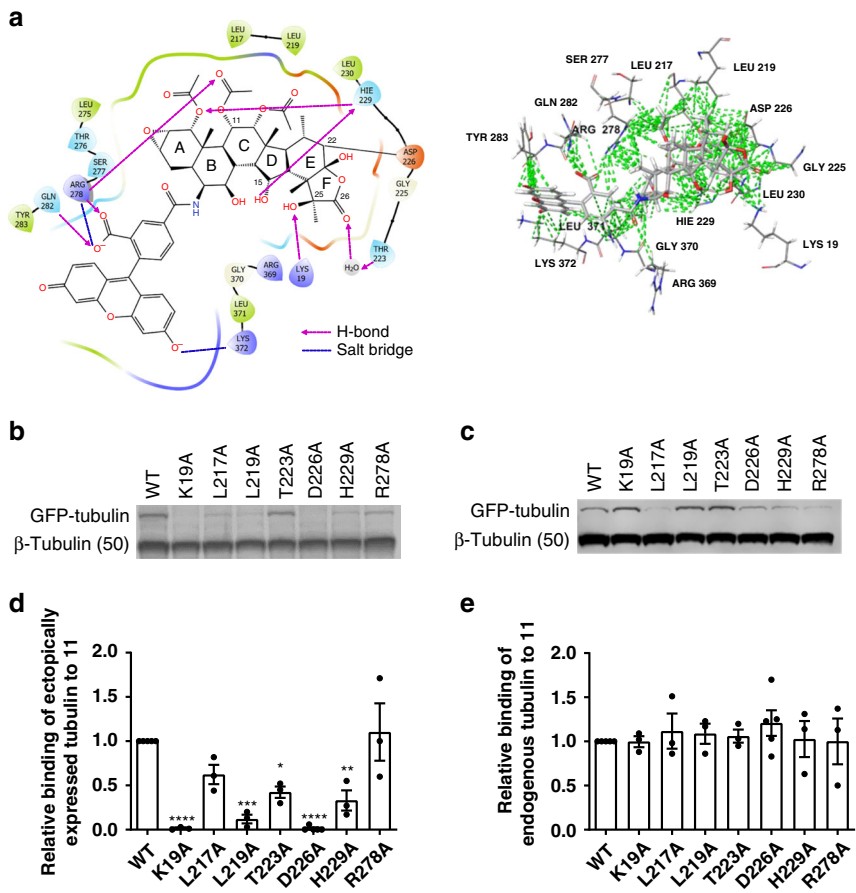

**Fig. 7 Systematic evaluation of β-tubulin residues that mediate taccalonolide binding. a** The key tubulin residues that mediate the binding affinity of **12** in the docking model structure generated by CovDock. The residues within the radius of 3.5 Å from the probe are displayed in the left graph and hydrophobic interactions are depicted by green dashed lines in the right graph. **b**, **c** HeLa cells were transfected with GFP-tagged TUBB1 constructs with indicated mutations then treated with 1 μM **11** for 8 h. Probe-treated cells were lysed and subjected to immunoblotting. **b** An anti-fluorescein antibody was used to detect **11** bound to endogenous β-tubulin (lower band, 50 kDa) and the GFP-tubulin constructs (upper band, 77 kDa). **c** β-tubulin immunoblotting showed the expression of the GFP-tubulin constructs (upper band, 77 kDa) and endogenous tubulin (lower band, 50 kDa). **d**, **e** Ratio of the binding of **11** to each ectopically expressed tubulin mutant **d** or endogenously expressed tubulin **e** normalized to wild type (WT). Data are shown as average ± SEM for $n = 3$ independent experiments other than WT and D226A, which are from $n = 5$ independent experiments. One-way ANOVA and Tukey's post-hoc test were used to calculate statistical significance between each condition. Significance as compared to the R278A mutant that did not impact binding but was not used in data normalization is shown: $*p < 0.05$, $**p < 0.01$, $***p < 0.001$, $****p < 0.0001$. Source data are provided as a Source Data file.

cells that expressed wild type or mutant GFP-tubulin constructs were treated with **11** at 1 μM for 8 h followed by immunoblotting using anti-β-tubulin and anti-fluorescein antibodies. In the β-tubulin immunoblot, bands were detected for each of the GFP-tagged forms of β-tubulin (wild type, D226N, or D226A) at 77 kDa that were distinct from endogenous β-tubulin (50 kDa) (Fig. 6d). A fluorescein immunoblot of these same samples demonstrated that **11** was able to interact with the ectopically expressed wild type form of GFP-tubulin, but not either D226 mutant, although it interacted with the endogenously expressed tubulin for all cases as an internal control (Fig. 6e). These results confirm the critical role of the 22,23-epoxide and β-tubulin D226 in taccalonolide-tubulin binding[18].

Encouraged by the results of the D226 mutagenesis, additional β-tubulin mutants were constructed in a similar fashion. The analysis of the 5EZY crystal structure[18] and the covalent docking models of both **2** (Fig. 1e) and **12** (Fig. 7a) revealed a potential for interaction of the taccalonolide skeleton with 6 residues via H-bonds and/or salt bridges and hydrophobic interactions (i.e. H229, R278, K19, Q282, T223, and K372) and 9 additional β-tubulin residues via only hydrophobic interactions (i.e. L217, R369, L219, L230, L371, Y283, G225, G370, and S277). Overall,

the prediction of key interacting residues based on covalent docking to the 5EZY crystallographic structure qualitatively matched the molecular dynamics simulation results of **2** using the cryo-EM structure of a mammalian microtubule[26]. Thus, besides D226, 6 β-tubulin residues (i.e., K19, H229, R278, L217, L219, and T223) were selected for mutagenesis with similar procedures, as described above (Supplementary Table 9) followed by immunoblotting to determine their impact on taccalonolide binding. It is important to note that although some of these mutants were expressed at lower levels than wild type GFP-tubulin (Fig. 7b), they were each able to be incorporated into microtubules that were effectively stabilized by the addition of untagged taccalonolide AJ as determined by visualization of the GFP-tag (Supplementary Fig. 7). The binding ratio of the probe **11** to each GFP-tubulin mutant or endogenous β-tubulin was quantifed by a ratio of fluorescein signal (Fig. 7b) compared to the β-tubulin signal (Fig. 7c) for each band. The binding ratio for each mutant was normalized to that of wild type β-tubulin to determine the relative significance of each tested β-tubulin residue for probe binding to β-tubulin (Fig. 7d).

For all of the samples, the probe bound to the endogenous β-tubulin at a similar level (Fig. 7e) indicating the mutagenic

manipulations did not affect the extent of the intrinsic binding of the taccalonolide to β-tubulin. Further analysis of the relative binding ratios for GFP-tubulin (Fig. 7d) clearly showed that the mutation of different β-tubulin residues distinctly affected the binding of the probe to β-tubulin. The extent by which different β-tubulin residues affected taccalonolide binding correlated with the type of interaction (Fig. 7a and Supplementary Fig. 8), as well as the distance of the residue to the covalent binding site (C-22 of **11**) (Supplementary Table 11). Specifically, the β-tubulin residues K19, H229, and R278 were predicted to form both H-bonds and hydrophobic interactions with specific moieties on the probe (Fig. 7a), which were progressively distanced from the site of covalent binding (C-22 on probe ring E) (Supplementary Fig. 8a and Supplementary Table 11). The K19 side chain strongly interacted with several moieties on ring F (i.e. 25-OH, 26-CO, and 28-Me) which were all close to the covalent binding site (shortest distance, 3.3 Å from C-26 to C-22) (Fig. 7a). Accordingly, the K19A mutation almost completely abrogated the probe binding to a similar extent as the D226 mutations (Figs. 6e, 7b, d). In contrast, the H229 side chain mainly interacted with the moieties on rings B-D which were relatively remote from the covalent binding site (shortest distance, 15-OH, 5.2 Å to C-22) (Fig. 7a). Thus, the H229A mutation only moderately inhibited binding (Fig. 7b, d). This observation was consistent with previous SAR studies demonstrating that 25-OH esterification of the 22,23-epoxy taccalonolides dramatically suppressed anti-proliferative potency while modification of 15-OH only slightly affected potency[21]. Although R278 was predicted to interact with both the taccalonolide core structure (e.g. 11-Me on ring B and 11-Ac on ring C) and the fluorescein moiety via hydrophobic interactions, two H-bonds, and a salt bridge (Fig. 7a), the R278A mutation did not affect probe binding most likely due to the remote location of the interacting sites from the covalent binding site (e.g., 11-Ac, 8.8 Å to C-22). A similar trend was observed for L217 and L219, which were predicted to interact with the taccalonolide core structure via hydrophobic interactions (Supplementary Fig. 8c, d). Although both L217 and L219 strongly interacted with 11-Ac and 12-Ac on taccalonolide ring C, only L219 showed hydrophobic interactions with 21-Me that collocated with the binding site on ring E. Therefore, the L219A mutation significantly suppressed the probe binding, while the L217A mutation only exhibited a weak inhibitory effect (Fig. 7a, d). An exception was observed for T223, which was predicted to interact with 26-CO on ring F through a H-bond-connected $H_2O$ bridge in the 5EZY crystal structure (Figs. 1e and 7a). The T223 hydroxyl was also predicted to play an important role in fixing the carboxylate of D226 to facilitate the covalent reaction with the 22,23-epoxide on the basis of the molecular dynamics simulation of **2**[26]. Interestingly, the T223A mutation only had a moderate effect on binding (Fig. 7a, d) despite the predicted importance of this residue for taccalonolide binding. To gain additional insight into the impact of the T223A and H229A mutants on the rate of taccalonolide binding, we evaluated the binding of the probe **11** at 1, 2, 4, and 8 h. The extent of binding of **11** to endogenous tubulin was minimal at 1 h with increased binding observed at 2–4 h regardless of what form of tubulin (mutant or wild type) was ectopically expressed (Supplementary Fig. 9). Binding to ectopic wild type tubulin was detectable at 4–8 h with the T223A and H229A mutants showing delays in the rate and extent of binding with a more pronounced effect for the H229A mutant, consistent with Fig. 7 (Supplementary Fig. 9).

Together, the analysis above suggests that the structures of the taccalonolide rings E and F and their direct interactions with β-tubulin residues (e.g. K19 and L219) are critical for the correct positioning of the taccalonolide core structure into its binding pocket to facilitate the covalent reaction between D226 and the 22,23-epoxide. It is also reasonable to hypothesize that other β-tubulin residues, including L217, H229, and R278, that interact with the taccalonolides at sites relatively remote from the site of the covalent interaction (Fig. 7a) are less important for covalent binding but may play an important role in mediating the post-reaction stability of the taccalonolide-microtubule complex. Thus, these findings that empirically establish the critical β-tubulin residues and the potential pharmacophore of taccalonolides may provide important guidance for the rational design and generation of 'drug-like' taccalonolide analogues via strategies such as semi-synthesis and structural simplification[34].

**Flu-tacca-7 as a superior tubulin probe.** The Flu-tacca probes represent a class of irreversible, fluorogenic microtubule probes that can be utilized for cellular microtubule imaging and binding studies under conditions that have conventionally been unfavorable for the use of non-covalent probes. To demonstrate the utility of our cell-permeable probes for cellular imaging applications, we compared the cellular microtubule staining activities of Flu-tacca-7 (**11**) to Tubulin Tracker Green (Thermo Fisher Scientific, Oregon Green™ 488 Taxol, Bis-Acetate) and the far-red probe siR-Tubulin (Cytoskeleton), two commercial taxane-based probes commonly used for microtubule imaging in live cells. Tubulin Tracker Green was not amenable to visualization prior to washing excess probe from the media (Fig. 8a) and pluronic F-127 was necessary to facilitate drug loading into cells and decrease background fluorescence (Supplementary Fig. 10). In contrast, Flu-tacca-7 was effectively visualized without washing or the use of additional reagents to facilitate cell loading providing comparable convenience to the far-red probe siR-Tubulin (Fig. 8a). However, the intracellular staining of Flu-tacca-7 was superior to both Tubulin Tracker Green and siR-Tubulin when microtubules were depolymerized under chilled conditions or in cells expressing drug efflux transporters (Fig. 8). Brief chilling was sufficient to destabilize microtubules in the presence of the commercial taxane probes, resulting in loss of probe signal, which could only be detected slightly over background when images were hyper-contrasted (Supplementary Fig. 10). In contrast, visualization of the taccalonolide probe **11** was retained at a similar level before and after chilling due to the covalent and irreversible nature of its binding to β-tubulin (Fig. 8a). The tubulin labeling efficacy of **11**, Tubulin Tracker Green, and siR-Tubulin were also compared in SK-OV-3-MDR1-M6/6 ovarian cancer cells, which express 1000-fold increased levels of the P-glycoprotein drug efflux pump MDR-1 as compared to the parental SK-OV-3 cell line (Fig. 8b and Supplementary Fig. 11). While **11** was effectively visualized in SK-OV-3-MDR-1-M6/6 cells, cellular staining of either Tubulin Tracker Green or siR-Tubulin was barely detectable above background in these multi-drug resistant cells (Fig. 8b). This observation is consistent with the fact that siR-Tubulin is recommended to be used in combination with verapamil to block probe efflux as both it and Tubulin Tracker Green are P-glycoprotein substrates. Additionally, **11** retained potency, efficacy, and microtubule binding in a βIII-tubulin expressing HeLa cell line (Supplementary Fig. 12), representing another clinically relevant form of taxane drug resistance. Therefore, our Flu-tacca irreversible microtubule probes are superior to commercial taxane-based probes, combining a high degree of specificity, brightness, and convenience with their ability to be used under conventionally unfavorable conditions (i.e., chilling and

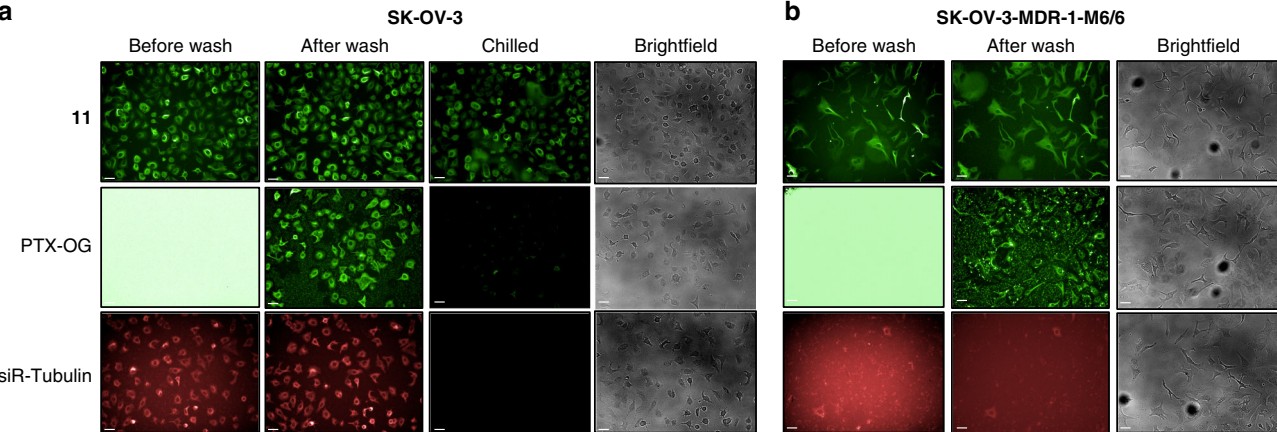

**Fig. 8 Taccalonolide probes compared to commercial taxane-based probes. a** SK-OV-3 human ovarian cancer cells were treated with 0.5 μM of **11**, Tubulin Tracker Green (PTX-OG; Thermo Fisher; middle), or siR-Tubulin (Cytoskeleton, Inc; bottom) for 5 h and imaged before treated media was removed (before wash), after the treated media were replaced by fresh medium (after wash), or after the treated cells were chilled at −20 °C for 20 min and fixed (chilled). **b** SK-OV-3-MDR1-M6/6 human ovarian cancer cells expressing P-glycoprotein were treated with 0.5 μM of each probe for 5 h and imaged at 37 °C before and after washing. The same image acquisition and visualization conditions were used for all images obtained for each probe. Brightfield images are shown in the far right column for each figure. Scale bars = 50 μm for all images. Source data are provided as a Source Data file.

drug-resistant cells) without the need for additional pharmacological manipulations.

## Discussion

While three distinct classes of covalent microtubule stabilizing agents (i.e., the taccalonolides, cyclostreptin, and dactylolide/zampanolide) have been evaluated for their potential as cancer therapeutics that circumvent taxane associated drug resistance in biochemical and cellular assays[32], the taccalonolides are the only class that has demonstrated in vivo efficacy in both drug sensitive and resistant tumor models[17,20,21,25]. Here we describe the development of a combinatorial chemical proteomics approach that enabled confirmation of the target specificity of taccalonolide binding to β-tubulin in cellular assays and identification of key structural features of both the taccalonolides and β-tubulin that are critical for drug–target binding. Our combinatorial strategy to empirically establish the key taccalonolide binding residues in cell-based assays should be broadly useful for detailed ligand-protein binding studies and structure-based optimization of quite a number of existing drugs and/or drug candidates that covalently modify their targets.

While the immunofluorescence and immunoblotting with the Flu-tacca probes clearly demonstrate that the predominant cellular target of the taccalonolides is indeed a covalent interaction with β-tubulin, we cannot rule out the possibility of other minor covalent or non-covalent targets. One important consideration is the finding that taccalonolides lacking the critical covalent binding moiety, the 22,23-epoxide, show no antiproliferative, cytotoxic, or microtubule bundling activities in cells and do not directly interact with tubulin in biochemical assays. The lack of any detectable target engagement or bioactivity of probes lacking this epoxide in the current study lends further support to the specificity of this interaction.

An unanticipated finding from the generation of the Flu-tacca probes in this study is the identification of a strategy to improve the binding affinity and microtubule stabilizing potency of the taccalonolides by targeting a binding pocket nearby the taccalonolide binding site that engages the M-loop of β-tubulin, which is integral in pharmacological microtubule stabilization[18,35]. These findings provide an opportunity to develop additional 'drug-like' taccalonolide analogues with sub-nanomolar cellular potency, which is desired for targeted drug delivery strategies (i.e., peptide-

and antibody-drug conjugates) and paves the way for further development of the taccalonolides as microtubule stabilizers for cancer therapy. A further evaluation of the critical taccalonolide moieties and β-tubulin residues that mediate binding and biological activity will continue to provide mechanistic insight to optimize our models of taccalonolide binding and design more optimal semi-synthetic and possibly fully-synthetic taccalonolide-like compounds that could have efficacy in drug resistant models due to their irreversible target binding. Additionally, the fluorogenic taccalonolide-based probes represent highly specific, irreversible tubulin-labeling probes that have superior utilities as compared to commercial taxane-based probes.

## Methods

**Synthesis of taccalonolide probes**. The detailed procedures for the synthesis of taccalonolide probes **3–12** are described in Supplementary Methods and reaction schemes provided in Supplementary Figs. 82–96.

**Computational modeling**. The computational modeling experiments were conducted using the Schrödinger Small-Molecule Drug Discovery Suite (2018-4). The crystal structure 5EZY was downloaded from PDB and optimized using the Protein Preparation Wizard following the standard protocol[36]. Briefly, the structure was first preprocessed by assigning bond orders, adding hydrogen atoms, creating zero-bond orders to metals, and creating disulfide bonds between two sulfur atoms within 3.2 Å from each other. Prime was used to predict and fill in the missing side chains. Water molecules beyond 5 Å from het groups were removed. To further optimize the model, the H-bond assignment was optimized by sampling water orientations and using PROPKA to assign protonation states of side chains at pH 7.0. All Asp, Glu, Arg, and Lys residues were left in their charged state and the proper His tautomer was also manually selected to maximize hydrogen bonding. Next, a brief relaxation was performed on the structure. This is a two-part procedure that consists of optimizing hydroxyl and thiol torsions in the first stage followed by an all-atom constrained minimization in the second stage to relieve clashes. The minimization was terminated when the RMSD reached a maximum value of 0.30 Å. The optimized protein structure was simplified by only retaining chain B comprising the taccalonolide AJ-β-tubulin complex and removing the other 5 chains. All water molecules were removed except for the one that formed H-bonds between β-tubulin T223 and the 26-carbonyl group of taccalonolide AJ.

The ligand structures were optimized using the Ligand Preparation Wizard (LigPrep)[36]. Briefly, energy minimization of the ligands was conducted using the OPLS3 force field. The ionization states were generated at pH 7.0 using Epik and the dominating tautomer of each ligand was retained for docking experiments.

Further covalent docking experiments were performed using CovDock[27]. In the CovDock wizard, D226 was selected as the reactive residue. The docking box was centered on the coordinates X 3.2/Y −63.5/Z 22.6 in the length of 20 Å. The covalent reaction was defined as an epoxide opening reaction that was constrained to take place only on C-22 of the ligands. The 'thorough' docking mode was used for the optimization of poses. The cutoff of 2.5 kcal/mol was applied for retaining

poses for further optimization in each cycle. The top 10 low-energy poses were generated and retained for each docking experiment. All the 10 docking models were visually checked for the binding interactions of the taccalonolide core structure to filter out the inappropriate binding models with the taccalonolide core structures that were significantly rotated or positioned outside the binding pocket. The lowest-energy pose showing correct spatial arrangement of the taccalonolide core structure was selected for analysis of the ligand-protein binding modes.

**Cell lines.** HCC1806 (CRL-2335) and HCC1937 (CRL-2336) human triple-negative breast cancer cells, HeLa (CCL-2) cervical cancer cells and SK-OV-3 (HTB-77) ovarian cancer cells were obtained from ATCC (Manassas, VA) and validated by STR profiling (Genetica). SK-OV-3 cells stably overexpressing Pgp by adenoviral-mediated expression of MDR1 were obtained from Dr. Susan Kane and subcloned by limiting dilution to isolate the single-cell clones utilized in these studies as SK-OV-3-MDR-1-6/6[20]. A single-cell clone from transfection of HeLa cells with βIII-tubulin, designated wild type βIII, was constructed and obtained from Dr. Richard Ludueña[20]. HCC1806 and HCC1937 cells were cultured in RPMI 1640 media (Corning) with 10% FBS (Cellgro) and 50 μg/mL gentamicin (Gibco). HeLa, βIII-tubulin expressing HeLa, SK-OV-3 and SK-OV-3/MDR-1-6/6 cells were grown in BME media with Earle's salts (Gibco) with 10% FBS, 1× final 1% Glu-taMax™ Supplement (Gibco), and 50 μg/mL gentamicin. The use of HeLa cells allows for the direct comparison of the in vitro potency as compared to other compounds of this class, which have been reported predominantly in this line and compound potencies are consistent among three additional cancer cell lines. HeLa cells were also used for ectopic expression of tubulin mutants due to the high degree of transfectability of this cell line. Cells were tested for mycoplasma contamination using the Mycoplasma Detection Kit-Quick Test (Cat: B39032, Lot: JW004).

**Antiproliferative assay.** The sulforhodamine B (SRB) assay was utilized to examine the antiproliferative and cytotoxic effects of the compounds[37,38]. Approximately 2000 cells per well (for SK-OV-3, HeLa and βIII-tubulin expressing HeLa) or 4000 cells per well (for HCC1806 and HCC1937) were seeded in 96-well plates. For each biological replicate, cells were seeded in triplicate with each concentration of compound or vehicle control for 48 h in a final volume of 200 μL. The plates were fixed with 10% trichloroacetic acid for protein precipitation of adherent cells and then washed with distilled water. In total 100 μL of SRB dye, which binds protein stoichiometrically, was added and then unbound dye removed with 1% acetic acid followed by the addition of 200 μL 10 mM Tris to to solubilize the dye, which was quantified by absorbance at 560 nm. The percent growth of treated cells relative to the density at the time of drug addition was calculated as compared to vehicle treated cells. Concentration-response curves were generated by non-linear regression analysis using Prism software 7.04 (GraphPad) and the $GI_{50}$ of each compound was calculated and defined as the concentration that caused a 50% decrease in cellular proliferation in the 48 h of drug incubation in comparison to vehicle control from 3 independent experiments.

**Live cell fluorescence imaging and immunofluorescence.** HCC1937, SK-OV-3, and SK-OV-3/MDR-1-6/6 cells were plated in PerkinElmer cell carrier imaging 96-well plates at a density of 8000–10,000 cells/well. HeLa and HeLa βIII-tubulin overexpressing human cervical cancer cells were plated in PerkinElmer cell carrier imaging 96-well plates at a density of 4000 cells/well. Cells were treated with vehicle control or compounds at the indicated final concentration for each individual experiment. Tubulin Tracker Green and siR-Tubulin stock solutions were prepared in anhydrous DMSO (Sigma Aldrich) at concentrations of 2 mM or 1 mM, respectively. Pluronic® F-127 (Invitrogen) was added at a 1:1 ratio from a 20% (w/v) DMSO stock solution where indicated. For live cell imaging, cells were imaged 5 h after treatment with compound and then washed with fresh media or Hank's Balanced Salt Solution (HBSS) (Sigma-Aldrich) supplemented with 2 mM CaCl₂ and 0.8 mM MgSO₄ and imaged on the Operetta high content imager using Harmony software (PerkinElmer). HeLa and HeLa βIII-tubulin overexpressing human cervical cancer cells were treated with vehicle (ethanol), 0.5 μM or 5 μM of probes respectively for 5 h treatment in HBSS. Wells were washed prior to fixing with methanol. Images were taken with the Operetta at ×20. For colocalization experiments, cells were fixed with methanol after treatment and subjected to immunofluorescence for β-tubulin at 1:1000 (Sigma T-4026) with goat anti-mouse IgG (H + L) cross-absorbed secondary antibody, Texas Red-X at 1:200 (Invitrogen T-862), while the fluorescein-tagged taccalonolide was directly detected. Probe treated SK-OV-3 cells were imaged in medium prior to wash or in PBS after washing or chilling at −20 °C for 20 min and fixed with methanol. For confocal imaging, HCC1937 cells were treated with 0.05–5 μM taccalonolide probes for 6–24 h on glass coverslips in a 6-well plate and fixed with methanol prior to β-tubulin immunofluorescence at 1:1000 (Sigma T-4026) using goat anti-mouse IgG (H + L) cross-absorbed secondary antibody, Texas Red-X at 1:200 (Invitrogen T-862). Confocal images were acquired using a SP8 Leica DMi8 microscope using a ×63 oil objective. All images are representative of the phenotypes observed from examining multiple fields from at least 2 independent experiments.

**Tubulin polymerization assay.** Biochemical tubulin polymerization assays were performed using purified porcine brain tubulin (Cytoskeleton). In individual wells of a 96-well plate, 1 μL of each ×100 drug stock was incubated with 20 μM porcine brain tubulin in GPEM glycerol buffer (1 mM GTP, 10% glycerol, 80 mM PIPES pH 6.9, 2 mM MgCl₂ and 0.5 mM EGTA) in a final volume of 100 μL. Pure porcine tubulin was prepared on ice at 4 °C to inhibit tubulin polymerization until the assay was initiated, while the plate reader was pre-warmed to 37 °C. Tubulin polymerization was measured every minute for an hour by light scattering at 340 nm in a Spectramax plate reader using SoftMax software (Molecular Devices). Light scattering was normalized to the initial measurement for each well. For the probe-tubulin binding assay, samples were prepared on ice in tubes instead of a 96 well plate, and moved to a 37 °C heat block to initiate binding and polymerization. The time zero (0') sample consisted of tubulin polymerization buffer prior to addition of taccalonolide probe. At each designated time point, 2 μL of the sample was added to 50 μL NuPAGE sample buffer with 20% β-Mercaptoethanol and 10% of the resulting sample was subjected to PAGE and immunoblotted for β-tubulin at 1:1000 (abcam, ab6046) or fluorescein at 1:500 (abcam, ab19491) with IRDye 680 or 800 goat anti-rabbit secondary antibodies at 1:10,000 (LI-COR Biosciences) and imaged on an Odyssey FC (LI-COR Biosciences).

**Site-directed mutagenesis.** The QuikChange II XL Site-Directed Mutagenesis Kit (Agilent Technologies) was used according to the manufacturer's directions with any changes noted. The template for mutagenesis was the human TUBB1 ORF mammalian expression plasmid, C-GFPSpark tag from Sino Biological Inc. (HG11626-ACG) using the primers listed in Supplementary Table 9. After Dpn I digestion, amplification products were stored at 4 °C until transformation into DH10B or XL10-Gold competent cells. DNA constructs were isolated using the QIAGEN Plasmid Midi Kit and Thermo Scientific GeneJet Plasmid Mini Kit. DNA concentrations were measured using a NanoDrop 2000 (Thermo Fisher Scientific). All constructs were sequenced using GENEWIZ and sequences verified using SnapGene.

**Activity-based protein profiling and immunoblotting.** For cellular binding studies, HeLa cells were seeded to 80–90% confluence in 6-well dishes and transiently transfected with the wild type or mutant tubulin constructs using Lipofectamine 3000 Transfection Reagent (Thermo Fisher Scientific) for 16-18 h. Media with the lipofectamine reagent were removed from the wells and replaced with fresh BME media for approximately 24 h prior to drug treatment. Cells were treated with 1 μM 11, 1 μM taccalonolide AJ or EtOH vehicle for 1–8 h, respectively. Cells were collected by scraping with a cell lifter and lysed with cell extraction buffer (Invitrogen) supplemented with protease inhibitor cocktail (Sigma-Aldrich), 50 mM NaF, 200 μM Na₃VO₄ (Sigma-Aldrich), and 1 mM phenylmethylsulfonyl fluoride (PMSF) (Sigma-Aldrich). Protein concentration was determined by a Coomassie Plus assay kit (Thermo Scientific), equal amounts of protein resolved by SDS–PAGE on NuPage Bolt 10% Bis-Tris gels (Life Technologies), and transferred to Immobilion-FL PVDF membranes (Millipore). Membranes were blocked in Odyssey blocking buffer (LI-COR Biosciences, Lincoln, NE, USA) and probed with anti-fluorescein at 1:500 (abcam, ab19491) or β-tubulin 1:1000 (abcam, ab6046) with IRDye 680 or 800 goat anti-rabbit secondary antibodies at 1:10,000 (LI-COR Biosciences, T8660) and imaged on an Odyssey FC (LI-COR Biosciences). βIII-tubulin was detected using a monoclonal antibody produced in mouse (1:400) (Sigma-Aldrich) clone SDL.3D10, ascites fluid.

Revert total protein staining was utilized to demonstrate relative equal total protein for each lysate (LI-COR Biosciences). The relative binding ratio of mutants was calculated as: (fluorescein signal$_{mutant}$/tubulin signal$_{mutant}$)/(fluorescein signal$_{wildtype}$/tubulin$_{wildtype}$) and expressed as percent of the wildtype signal for each independent experiment. For imaging studies, wild type or mutant TUBB1-GFP constructs were transfected into HeLa cells (40,000 cells/well in a 96-well plate) using Lipofectamine 3000 for 16–18 h prior to washing and replacing with fresh media. After 7 h recovery, cells were imaged before and after treatment with vehicle or 100 nM taccalonolide AJ for 22 h using the Operetta. Uncropped blots can be found in the source data file.

**Cellular biotransformation assays.** HCC1937 cells were grown to 90% confluence then treated with vehicle (ethanol), or 1 μM 10 for 8 h in a total volume of 5 mL. The media was harvested while the cell pellet was lysed by dounce homogenization in a hypotonic buffer (1 mM EGTA and 1 mM MgSO4, pH 7) after 2 washes with 1× PBS. The cell lysates were extracted by ethyl acetate. The organic layers were dried down and re-dissolved in MeOH for LCMS analysis.

**qRT-PCR.** RNA was isolated from SK-OV-3 and SK-OV-3/MDR-1-6/6 ovarian cancer cells by Trizol and chloroform extraction. The RNA pellet was resuspended in nuclease-free water and quantified using a Nanodrop 2000. RNA was converted to cDNA with iScript Reverse Transcription Supermix for RT-qPCR (Bio-Rad) and qRT–PCR was completed using iTaq Universal SYBr Green Supermix (Bio-Rad). Human Pgp primers (Sigma-Aldrich) were generated based on previous reports[39] as Pgp1: 5′- AAAGCGACTGAATGTTCAGTGG-3′ and Pgp2: 5′- AATAGATG CCTTTCTGTGCCAG-3′ and specificity was confirmed using NCBI Primer-BLAST. Human GAPDH primers: 5′- GCAAATTCCATGGCACCGT-3′ and

5′- TCGCCCCACTTGATTTTGG-3′. Relative hPgp mRNA transcript levels were evaluated and presented from two biologically independent experiments, each performed in duplicate.

**Statistics**. For binding studies, a one-way ANOVA with Tukey's multiple comparison post-hoc test and adjustment for multiple comparisons were used to determine statistical significance between each condition and significance of mutants as compared to wild type depicted in the figure. Exact n values and P values are in Supplementary Table 10. For qRT-PCR comparing human Pgp (hPgp) expression in SK-OV-3 and SK-OV-3/MDR-1-6/6 cell lines a one-tailed $t$-test was performed to give a P value of 0.0307.

**Reporting summary**. Further information on research design is available in the Nature Research Reporting Summary linked to this article.

## Data availability

The source data underlying Table 1, Figs. 4–8, and Supplementary Figs. 2–4, 6, 7, 9–12 are provided as a Source Data file. Chemical synthesis procedures and NMR data on all compounds can be found in Supplementary Figs. 13–96. The x-ray crystallographic coordinates for structures reported in this study have been deposited at the Cambridge Crystallographic Data Centre (CDCC) under deposition number 1907790. These data can be obtained free of charge from the Cambridge Crystallographic Data Centre via www.ccdc.cam.ac.uk/data_request/cif. The data underlying the findings of this study can be obtained from the corresponding authors upon reasonable request.

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

## Acknowledgements

This work was funded by R01 CA219948 to A.L.R. and L.D. We would like to thank Dr. Madesh Muniswamy (University of Texas Health Science Center San Antonio, TX) for use of the confocal microscope. We would also like to thank Dr. Lily Q. Dong (University of Texas Health Science Center San Antonio, TX) for the DH10B cells for site-directed mutagenesis and centrifuges required for the DNA midipreps.

## Author contributions

L.D. and A.L.R. conceived the project. L.D. performed the organic synthesis and computational modeling. S.S.Y. conducted the biological assays with the exclusion of the confocal imaging that was done by K.R. L.D. and A.L.R. wrote the manuscript.

## Competing interests

L.D., A.L.R., and S.S.Y. are listed as inventors on a pending patent application pertaining to the taccalonolide microtubule stabilizers. K.R. declares no competing interests.
