## [Peer Review File · Nature Communications]

Reviewers' comments:

Reviewer #1 (Remarks to the Author):

The manuscript by Du and coworkers describes the preparation of a series of taccalonolide fluorophores designed to probe the interactions between the drug and its biochemical target tubulin. The authors demonstrate that potent drug derivatives can be made, and that the agents behave in a manner similar to the parent drug. The taccalonolide derivatives are used in live cell imaging studies, and the molecular interactions with tubulin are probed with a series of mutated forms of tubulin. This is an interesting and novel series of investigations, that would, however, benefit from a few modifications.

1. The authors start with a series of acetyl esters that are rapidly hydrolyzed, and then focus on pivaloyl esters that are presumed to be more stable chemically. No consideration is given in the paper to the possibility that the enhanced activity of the pivaloyl esters is simply due to enhanced hydrophobicity and membrane permeability. This would be consistent with the data in Table 1, and particularly with the data shown in Figure 5. The difference between 2, 11, and 12 in 5c is small enough to be considered insignificant. The lack of any correlation between the data in Table 1 and Figure 5 underscores the possibility that permeability is an underlying cause for differences between the drugs. This needs to be addressed in detail.
2. There is no evidence presented anywhere in the paper that the pivaloyl esters are hydrolyzed by esterases. Hydrolysis is presumed, not shown. Statements are made regarding this issue with no supporting evidence.
3. The same is true for covalent binding to tubulin. The lack of activity of the 22,23-ene analogue 7 compared to the epoxy form does not provide support for such statements as made on lines 254, 258, and elsewhere about covalency. This statement is made elsewhere, and is not supported with actual demonstrations of the fact.
4. Figure 4c and d bottom and top panels need to be labeled (bright field...). Misspelling in Figure 4 legend, by the way.
5. Does the anti-fluorescein bind to the esters?

Reviewer #2 (Remarks to the Author):

Based on a recent X-ray crystal structure, and previously published fluorescent analogues of other tubulin-binding agents, Du, Risinger and coworkers describe the synthesis of the first highly cytotoxic fluorescent analogue of the taccalonolide family of natural products. This compound allowed the authors to demonstrate that these compounds modify tubulin covalently with exceptionally high specificity. It also provides the best covalent fluorescent probe of cellular tubulin reported to date. This study was very well conducted, the experiments are well controlled, and this manuscript can be published essentially as-is.

Reviewer #3 (Remarks to the Author):

Work on taccalonolides as irreversible tubulin binders has gained momentum in recent years. In the manuscript by Risinger et al, optimization of an already published taccalonolide prototype to yield better fluorescent probes was guided by results from a series of state-of-the-art cellular and biochemical experiments. Comparative evaluations were made with the untagged taccalonolide AJ and also with two commercial taxane-based probes. The novel Flu-tacca probes proved superior to the latter in terms of specificity, brightness, and cell spectrum (albeit Flutax-1 and Flutax-2 were not used in the comparison). In particular, they can stain taxane-resistant cells overexpressing efflux pumps although I am not aware they were tested on cells expressing beta-III tubulin. The experiments clearly support the view that the dipivaloyl protective groups serve two distinct roles for the taccalonolide probes: (i) effectively quenching extracellular fluorescence, and (ii) retaining the taccalonolides in a binding-deficient form prior to entry into cells where fluorescence and tubulin binding can both occur. As expected, the 22,23-ene analogues had no activity because of the crucial role of the epoxide ring in reactivity and covalent bond formation with the carboxylate of Asp226 (as known from earlier X-ray diffraction studies, i.e. PDB entry 5EZY). Site-directed mutagenesis experiments provided some new insights about how the side chains of seven selected

amino acids in the vicinity of the taxane-binding site, namely D226, K19, H229, R278, L217, L219, and T223, affected probe binding to beta-tubulin. All in all, the work reported in this paper is a significant contribution to the field, in my opinion.

I have only a few suggestions/recommendations for revision.

- The lack of binding to D226N and D226A variants cannot be said to be "intriguing" (line 272).
- 5EZY does not show the same conformation or protofilament arrangement that has been elucidated by cryo-EM studies on tubulin. Hence, any modeling results emanating from this structure have to be taken with care, given the very different lateral contacts, as discussed in reference 26, the results of which appear to have been underestimated and undervalued in the present manuscript.
- The computational modeling work, although certainly useful to provide some clues, appears to be rather naive. Most worrying is the fact that a single static structure was used throughout without taking into account that the (alpha-beta dimer)₂ present in the 5EZY X-ray crystal structure does not represent a "true" protofilament arrangement, as revealed in cryo-EM structures, for example. It would be interesting to show that the shape of the adjacent binding pocket that is proposed for the fluorescein moiety is maintained (or similar) in microtubules (e.g. DOI 10.1016/j.cell.2014.03.053). Another concern is the simple distance criterion on which selection of tubulin variants is based upon (Fig. 7A). One example is K19, which points to the hydroxyl on C25 although this H-bonding interaction is highly unlikely in an aqueous environment and the electron density is poor for the terminal amino group. An electrostatic interaction between the protonated amino group and the pentacyclic lactone oxygens, as analyzed in more detail in ref. 26, would appear more likely.
- Hydrophobicity, currently in S.I., deserves a more prominent place in the discussion as it would be expected to be a primary binding force for taccalonolides. In fact, an unexpected (and welcomed) result is the differential effects on binding suppression of L219A and L217A mutations.
- While CovDock is a useful computational tool to dock covalently bonding ligands into protein pockets, the retrospective finding that matches with the taccalonolide AJ pose present in the crystal structure of the complex can be found is not a big deal, especially because "The representative lowest-energy pose showing correct spatial arrangement of the taccalonolide core structure was selected for analysis of the ligand-protein binding modes."
- line 192: what is meant by "refined 5EZY structure"? Did the authors refine the structure deposited with the PDB or did they use the refinement provided as S.I. in ref. 26? The sentence in the Methods section "optimized using the Protein Preparation Wizard" is ambiguous and uninformative to user unfamiliar with Schrödinger's Small-Molecule Drug Discovery Suite.
- The moderate effect on probe binding of the T223A amino acid replacement is somewhat surprising, as stated. However, the sentence "this interaction may be less important for probe binding under physiological conditions than inferred from the crystal structure" (l. 330) is little informative. A role for the T223 hydroxyl (and some water molecules) in fixing the carboxylate of D226 has been proposed on the basis of molecular dynamics simulation results (ref. 26) but the possibility exists that another water molecule is effectively replacing this hydroxyl group. It would be interesting to know, perhaps in subsequent work, whether the binding kinetics are different for wild-type and T223A tubulin. A similar reasoning can be applied to the H229A variant since the structural and modeling data support the view that the imidazole ring of H229 provides a docking platform for this type of molecules.
- lines 336-338: the statement that L217, H229, and R278 "interact with the taccalonolides at sites relatively remote from the binding site" is at odds with the 5EZY structure and the depiction shown in Fig. 5e.
- lin 388: the statement "confirmation of the target specificity of taccalonolide binding to β -tubulin" appears to imply that this is the only target. How was this specificity assessed, if at all?

Lastly, since covalent stabilizers can partially overcome β III-tubulin-mediated drug resistance, it would have been nice to show that these fluorescent Flu-tacca probes, by virtue of the covalent bond they form with Asp226 of beta-tubulin, can also stain this drug-elusive isotype in cells. While a direct comparison is not deemed to be strictly necessary for publication of this report, some sentences in this regard might enrich the discussion. Incidentally, the section tagged "Discussion" really looks more like a "Conclusions" section.

We thank the reviewers for their thoughtful comments that we have used to make significant additions to the manuscript to improve its impact, clarity, and accessibility to readers. In particular we appreciate that all three reviewers found the initial submission to be “an interesting and novel series of investigations that would benefit from a few modifications”, “well conducted, well controlled and essentially published as-is” and “a significant contribution to the field (with) a few suggestions/recommendations for revision”, respectively. We have included point-by-point responses to the two reviewers who provided recommendations with significant additions made to the manuscript, including the addition of 3 new supplemental figures, underlined for reference.

Response to reviewer #1:

***Q1.** The authors start with a series of acetyl esters that are rapidly hydrolyzed, and then focus on pivaloyl esters that are presumed to be more stable chemically. No consideration is given in the paper to the possibility that the enhanced activity of the pivaloyl esters is simply due to enhanced hydrophobicity and membrane permeability. This would be consistent with the data in Table 1, and particularly with the data shown in Figure 5. The difference between 2, 11, and 12 in 5c is small enough to be considered insignificant. The lack of any correlation between the data in Table 1 and Figure 5 underscores the possibility that permeability is an underlying cause for differences between the drugs. This needs to be addressed in detail.*

A1. We agree that the difference between **2**, **11**, and **12** in the protein-based biochemical microtubule polymerization assay using pure tubulin in Figure 5c is not significant. However, **6** is clearly superior in potency to **8** in the biochemical assay but is less potent in cells, indicating that the increased affinity for polymerization of purified tubulin does not directly correlate with cellular activity. We further acknowledge that membrane permeability could be an important factor in determining the cellular potency of these compounds. However, the hydrophobicity of **8**, **9**, and **11** estimated by their cLogD values [**11** (6.18) > **8** (5.31) > **9** (5.07)] is not consistent with the relative potency of the three compounds as shown in Table 1 (**11** > **9** > **8**). Therefore, we are exploring whether factors, such as the stability of the amide linkages against intracellular hydrolysis, the sensitivity of the compounds to facilitated import and cell efflux mechanisms, and the binding efficiency and affinity of the deprotected probes to tubulin would all contribute to the effects of the probes on microtubule stabilization and antiproliferative potencies. Indeed, we found that the diacetyl protected compound **5** was rapidly (<1 h) and completely hydrolyzed in cell culture media to form the unprotected, charged form **6**. This data has been added to expand supplementary Figure 1. In contrast, the dipivaloyl analogues **8** and **11** were resistant to hydrolysis under the same condition even after extended incubation (16 h). We will continue to explore these mechanisms in future studies.

***Q2.** There is no evidence presented anywhere in the paper that the pivaloyl esters are hydrolyzed by esterases. Hydrolysis is presumed, not shown. Statements are made regarding this issue with no supporting evidence.*

A2. To directly address the question of cellular hydrolysis of the pivaloyl esters, we utilized the non-epoxidated 22,23-ene analogue **10**, which does not covalently bind microtubules, to facilitate the isolation of biotransformed material from cells. HCC1937 cells were treated with **10** for 8 h after which they were thoroughly washed with PBS and lysed under non-denaturing conditions. Compounds were isolated from cell lysates by EtOAc extraction and identified by LCMS analysis. We hypothesized that the pivaloyl esters on **10** would be hydrolyzed by intracellular esterases, resulted in the accumulation of the deprotected form, namely **10'**, in cells. The results, which are

now presented in supplemental figure 5, showed that **10** was indeed partially hydrolyzed to form **10'** in cells. Given the ubiquity of esterases and their known reactivity with these type of modifications, it is reasonable to hypothesize that the hydrolysis catalyzed by ubiquitous esterase would be a major deprotection mechanism for the pivaloyl-protected probes. However, we have modified our language to state that cellular hydrolysis occurs based on the new supplementary figure and only suggest that cellular esterases are a likely candidate for this hydrolysis.

Q3. The same is true for covalent binding to tubulin. The lack of activity of the 22,23-ene analogue 7 compared to the epoxy form does not provide support for such statements as made on lines 254, 258, and elsewhere about covalency. This statement is made elsewhere, and is not supported with actual demonstrations of the fact.

A3. The covalent binding of the 22,23-epoxy analogues **11** and **8** can be evidenced by the detection of the taccalonolide probe bound to tubulin on an immunoblot (Figures 6 and 7 and supplemental figure 5) of lysates that were prepared under denaturing and reducing conditions that would disrupt any non-covalent linkage. We have added language to clarify this in the manuscript.

Q4. Figure 4c and d bottom and top panels need to be labeled (bright field...). Misspelling in Figure 4 legend, by the way.

A4. The bright field images are now indicated in the legend.

Q5. Does the anti-fluorescein bind to the esters?

A5. We were not able to detect fluorescein-tacca-labeled tubulin by immunoblotting using the ester probe, **3**, which we attribute to cleavage by cellular esterases and is the rationale for the introduction of amide-based linkers in our current study.

Response to reviewer #3:

Q1. The lack of binding to D226N and D226A variants cannot be said to be "intriguing" (line 272).

A1. The term "intriguing" was deleted.

Q2. 5EZY does not show the same conformation or protofilament arrangement that has been elucidated by cryo-EM studies on tubulin. Hence, any modeling results emanating from this structure have to be taken with care, given the very different lateral contacts, as discussed in reference 26, the results of which appear to have been underestimated and undervalued in the present manuscript.

A2. We acknowledge that cryo-EM has an advantage in structural biology as it is believed to maintain the closer-to-native state due to the rapid freeze treatment of the sample. Though 5EZY does not show the same protofilament arrangement that has been elucidated by cryo-EM studies

on microtubules stabilized by other drugs, the recently reported cryo-EM structures for Taxol-, zampanolide- and peloruside-bound MTs have confirmed that the site and mode of binding described in the previous crystallographic studies are the same in the context of the assembled MT (J Mol Biol 2017, 429, 633–646; Current Opinion in Structural Biology 2017, 46, 65–70). This agreement is also supported by the molecular dynamics simulation results of zampanolide as described in Ref 26. Furthermore, the best resolutions of cryo-EM structures of MTs bound with taxane-site stabilizers are only at around 3.5–4.0 Å. At this resolution the side-chains are hard to trace and the individual interactions that stabilize the protein and contribute to ligand binding are difficult to discern. In contrast, the 5EZY structure provides atomic resolution at 2.05 Å and allows unambiguous determination of the orientation and conformation of the ligand and binding pocket residues. Importantly, recent studies on cryo-EM and X-Ray Crystallography indicate the two appear more as complementary techniques for comprehensive structural biology studies (Acta Crystallogr F Struct Biol Commun. 2017 Apr 1;73(Pt 4):174-183). Thus, it would be highly interesting, for future studies, to continue investigating the impact of the taccalonolides on MT structure and conformation using cryo-EM to further clarify the mechanism of action of this class of novel microtubule stabilizers collectively with other biochemical technologies.

Importantly, the 5EZY structure previously provided evidence for the unique mode of action of taccalonolide AJ, involving inhibition of GTP hydrolysis in the E-site of β -tubulin (Nat Commun. 2017 Jun 6;8:15787), which was confirmed by biochemical analysis. We have highlighted the fact that we used this structure as a starting point to predict the taccalonolide binding pocket and identify residues that may have an impact in our biochemical evaluations.

Ref 26 certainly provides useful methodology and guidance for improving our modeling protocols in future studies in order to more concisely and deeply understand the mode of action of taccalonolides and we have included additional references to this paper in our revised manuscript.

Q3. The computational modeling work, although certainly useful to provide some clues, appears to be rather naive. Most worrying is the fact that a single static structure was used throughout without taking into account that the (α - β dimer)₂ present in the 5EZY X-ray crystal structure does not represent a "true" protofilament arrangement, as revealed in cryo-EM structures, for example. It would be interesting to show that the shape of the adjacent binding pocket that is proposed for the fluorescein moiety is maintained (or similar) in microtubules (e.g. DOI 10.1016/j.cell.2014.03.053). Another concern is the simple distance criterion on which selection of tubulin variants is based upon (Fig. 7A). One example is K19, which points to the hydroxyl on C25 although this H-bonding interaction is highly unlikely in an aqueous environment and the electron density is poor for the terminal amino group. An electrostatic interaction between the protonated amino group and the pentacyclic lactone oxygens, as analyzed in more detail in ref. 26, would appear more likely.

A3. We agree with the reviewer that our modeling is not a major aspect of this manuscript and have added language that it's inclusion was simply used to qualitatively identify/predict key interactions between taccalonolide and tubulin residues within the binding pocket to provide guidance and support for the biochemical experiments and was used in concert with the more sophisticated modeling studies presented in ref 26.

However, in order to alleviate the reviewer's concern about docking using static receptor structure, we also took the receptor flexibility into account by performing Induced fit docking (IFD). The IFD protocol (*J. Med. Chem.* **2006**, *49*, 534-553) has been optimized and modularized in the Schrodinger computational platform. Briefly, IFD uses the docking program Glide to account for ligand flexibility and the Refinement module in the Prime program to account for receptor flexibility. The side-chain degrees of freedom in the receptor are sampled while allowing minor backbone movements through minimization. In this study, we first docked the structures of **6**, **8**, **11**, and **12** into the taccalonolide AJ binding pocket on 5EZY B-chain to generate pre-covalent complexes. For each modeling experiment, 200 ligand poses were sampled by Glide and the binding pocket residues within 5 Å to each ligand were refined by Prime. The resulted models were superimposed with the original 5EZY structure and visually checked for the ligand conformation and the ASP226(O⁻)-C22 distance. The lowest-energy models with optimal ligand conformation and ASP226(O⁻)-C22 distance was used as template for covalent docking of the individual ligand by CovDock to complete the covalent reaction.

Docking structures of β -tubulin-12 complex

Docking structures of β -tubulin-6 complex

White: CovDock
Green: IFD+CovDock

	12	12	6	6
	(CovDock)	(IFD+CovDock)	(CovDock)	(IFD+CovDock)
RMSD (Å, ligand)		0.62		1.46
RMSD (Å, all)		1.39		3.18
No. of H-bond	7	4	7	5
Hydrophobic interactions (No. of residues)	15	15	17	17

The initial IFD modeling of the deprotected probes **6** and **12** into the 5EZY structure generated lowest-energy tubulin complexes with ASP226(O⁻)-C22 distance at 3.1 and 2.9 Å, respectively, which is close to the optimal distance estimated for a nucleophilic reaction of the type studied here (2.9 Å) (*Adv Chem Ser* 1987, 215, 209–218.). In contrast, the IFD of the dipivaloyl protected probes **8** and **11** failed to generate any low-energy models with optimal ASP226(O⁻)-C22 distance indicating they are poorly fit into the binding pocket and are not prone to covalently bind ASP226 even though the receptor flexibility has already been taken into account. The structures of **6** and **12** were then covalently docked into their individual lowest-energy IFD model, respectively, using CovDock. The lowest-energy IFD-CovDock models (green) were then compared with the original lowest-energy CovDock models (white) as displayed in the above figure. Overall, the docking models generated by Covdock and IFD-CovDock showed high similarity suggesting both **12** and **6** are ideal ligands that fit well into the binding pockets without

significant clash with tubulin residues. For the modeling of **12**, the two docking models are almost identical with RMSD at 0.62 for the ligand and RMSD at 1.39 for the overall protein complex. Only minor movement of the binding pocket backbone was observed. In the case of **6**, slightly higher RMSD values were obtained (1.46 for the ligand and 3.18 for the protein complex) due to the higher flexibility of the β -alanine linker in this ligand's structure. The overall conformation of the binding pocket backbone is also very similar in the two models. For both ligands, the modeling with the two methods consistently predicted the hydrophobic interactions. However, the CovDock protocol consistently predicted more H-bond interactions than the IFD-CovDock protocol indicating the later one might have overestimated the flexibility of the binding pocket residues. While there is certainly room to further improve the IFD-CovDock protocol, the comparison of the modeling results indicated the original CovDock protocol already provided reasonable qualitative docking models for the purposes of our study. And importantly, follow-up studies using the same CovDock protocol have led to the successful prediction and generation of several new taccalonolide analogues with improved binding affinity to tubulin as evidenced by biochemical analysis, indicating that this methodology has real-life predictive value.

Regarding the H-bonding of K19, its amino group stayed in its protonated state in our modeling listed in Supplementary Table S11. In our model, the shortest distances from the amino protons of K19 to 25-OH and the 26-CO are 2.6 Å and 3.4 Å, respectively. Thus, it is more likely for the charged K19 to have strong electrostatic interaction with 25-OH other than 26-CO as predicted by Schrodinger.

***Q4.** Hydrophobicity, currently in S.I., deserves a more prominent place in the discussion as it would be expected to be a primary binding force for taccanolides. In fact, an unexpected (and welcomed) result is the differential effects on binding suppression of L219A and L217A mutations.*

A4. We have added a new image into Figure 7a displaying the 15 key β -tubulin residues that were predicted to interact with the taccalonolide probe via hydrophobic interactions. Overall, the prediction based on covalent docking matched the molecular dynamics simulation results as described in ref 26 in a qualitative way. The same 12 out of 15 residues were predicted by both approaches. We have added more details regarding hydrophobicity into the context and pointed out the consistency of the prediction results by the two approaches.

***Q5.** While CovDock is a useful computational tool to dock covalently bonding ligands into protein pockets, the retrospective finding that matches with the taccalonolide AJ pose present in the crystal structure of the complex can be found is not a big deal, especially because "The representative lowest-energy pose showing correct spatial arrangement of the taccalonolide core structure was selected for analysis of the ligand-protein binding modes."*

A5. As no docking constrains were defined, the CovDock protocol samples all possible ligand poses that would have an optimal ASP226(O⁻)-C22 distance and forces the covalent bond to form. This protocol provides relatively more comprehensive ligand sampling without predefined bias, but it can't distinguish a "good-fit model" from a "bad-fit model" in terms of the correct spatial arrangement of the taccalonolide core structure in the binding pocket. Thus, the top 10 low-energy docking models were retained from each docking experiment and visually checked for the binding interactions of the taccalonolide core structure to filter out the inappropriate binding models with the taccalonolide core structures that were significantly rotated or positioned outside the binding pocket. The tendency of each ligand to adopt "appropriate" conformation in the binding pocket was hypothesized to reflect the ease of the ligand to bind to β -tubulin without

significant clash with binding pocket residues. The analysis of the binding tendency was qualitatively consistent with the biochemical data obtained from the tubulin polymerization assay. We have added more details into the related sections of the manuscript to clarify these points.

Q6. *line 192: what is meant by "refined 5EZY structure"? Did the authors refine the structure deposited with the PDB or did they use the refinement provided as S.I. in ref. 26? The sentence in the Methods section "optimized using the Protein Preparation Wizard" is ambiguous and uninformative to user unfamiliar with Schrödinger's Small-Molecule Drug Discovery Suite.*

A6. The 5EZY structure was refined directly using the Protein Preparation Wizard which is modularized in the Schrödinger platform. We have included a reference 36 that details the methodology and protocol of the Protein Preparation Wizard and also added a brief description into the "Computational modeling".

Q7. *The moderate effect on probe binding of the T223A amino acid replacement is somewhat surprising, as stated. However, the sentence "this interaction may be less important for probe binding under physiological conditions than inferred from the crystal structure" (l. 330) is little informative. A role for the T223 hydroxyl (and some water molecules) in fixing the carboxylate of D226 has been proposed on the basis of molecular dynamics simulation results (ref. 26) but the possibility exists that another water molecule is effectively replacing this hydroxyl group. It would be interesting to know, perhaps in subsequent work, whether the binding kinetics are different for wildtype and T223A tubulin. A similar reasoning can be applied to the H229A variant since the structural and modeling data support the view that the imidazole ring of H229 provides a docking platform for this type of molecules.*

A7. We have added an additional supplementary figure evaluating the kinetics of binding between the taccalonolide probe and the T223A and H229A β -tubulin mutants as compared to wild type β -tubulin. The extent of binding of **11** to endogenous tubulin was minimal at 1 h with increased binding observed at 2 – 4 h regardless of what form of tubulin was ectopically expressed (Supplementary Fig. 9). Binding to ectopic wild type tubulin was detectable at 4 – 8 h with the T223A and H229 mutants showing delays in the rate and extent of binding with a more pronounced effect for the latter, consistent with Figure 7 (Supplementary Fig. 9).

Q8. *lines 336-338: the statement that L217, H229, and R278 "interact with the taccalonolides at sites relatively remote from the binding site" is at odds with the 5EZY structure and the depiction shown in Fig. 5e.*

A8. We have now included/modified two figures (Fig. 7a and Supplementary Fig. 8) and a table (Supplementary Table 11) to depict the interactions between the taccalonolide probe moieties and specific β -tubulin residues. Information has also been included to correlate the specific interactions with the importance of individual residues affecting taccalonolide binding.

Q9. *line388: the statement "confirmation of the target specificity of taccalonolide binding to β -tubulin" appears to imply that this is the only target. How was this specificity assessed, if at all?*

A9. While our experiments suggest a high degree of specificity for tubulin binding due to a lack of additional bands on the anti-fluorescein immunoblot and high colocalization with β -tubulin

immunostaining, we agree that these data do not demonstrate that there are no other targets, just that β -tubulin is the major cellular target. We have added language to clarify this point.

***Q10.** Lastly, since covalent stabilizers can partially overcome β III-tubulin-mediated drug resistance, it would have been nice to show that these fluorescent Flu-tacca probes, by virtue of the covalent bond they form with Asp226 of beta-tubulin, can also stain this drug-elusive isotype in cells. While a direct comparison is not deemed to be strictly necessary for publication of this report, some sentences in this regard might enrich the discussion. Incidentally, the section tagged "Discussion" really looks more like a "Conclusions" section.*

A10. We have added an additional figure, supplementary figure 12, demonstrating the antiproliferative efficacy and retention of microtubule bundling and staining by the probe **11** in HeLa cells that overexpress the β III-isotype of tubulin to demonstrate the efficacy of the taccalonolide probes in this additional taxane-resistant model.

Reviewers' comments:

Reviewer #1 (Remarks to the Author):

The authors largely addressed the issues raised. However, the new data in Supplementary Figure 5 seem rather poor and uncontrolled. The amount of time for analysis is not mentioned, there is no $t = 0$ time point, and there is no indication of what percentage conversion the mass spec peaks might represent, given potentially different ionization potentials.

Reviewer #3 (Remarks to the Author):

Overall, the authors made a great effort to address all the issues raised by the reviewers and the revised version has improved its clarity and impact.

However, I would like to add, for the sake of accuracy, a few comments/clarifications that may lead to very minor additional changes.

The first one refers to A2 in their response to reviewer #3's Q2. We are not talking here about the nominal resolution of an X-ray crystal structure versus that of a cryo-EM structure. What the reviewer was referring to was that in the 5EZY structure (which indeed "provides atomic resolution at 2.05 Å and allows unambiguous determination of the orientation and conformation of the ligand and binding pocket residues") some of the loops in beta-tubulin making up the binding site (and the M-loop in particular) have a conformation that is (necessarily) different from that known to exist in microtubules (cf. Fig. 2 [<https://link.springer.com/article/10.1007%2Fs10822-019-00208-w>] and Fig. S1 in ref. 26 [https://static-content.springer.com/esm/art%3A10.1007%2Fs10822-019-00208-w/MediaObjects/10822_2019_208_MOESM7_ESM.docx]). For this reason, since the fluorescent moiety is (very reasonably) proposed to interact with R278 and Q282 in the M-loop (Fig. 5e-h), the same docking experiment could have been done with PDB entries 3J6G ("optimized" by using restrained molecular dynamics simulations in D. Baker's group) or 5SYG (better nominal resolution but presenting gross steric clashes) [DOI: 10.1016/j.jmb.2017.01.001]. No further action is required.

The second one is related to Q6 and A6 in their answer to the same reviewer. The terms "refine" and "refinement" are widely used by X-ray crystallographers in a very precise context; molecular modelers also "refine" a structure when they remove steric clash and optimize interactions during an energy minimization. What was done here was to "prepare" the coordinates deposited with the PDB to carry out some molecular modeling tasks. For this reason I believed -and still believe- that the use of "refine" is somewhat inappropriate and misleading and therefore should be avoided. The text now reads "optimized" (l. 468) and this is entirely acceptable and later on "To further refine the structure, ..." (l. 473); in the present context I do not see an objection.

Lastly, and in relation to Q8/A8, I still do not understand the sentence "...L217, H229, and R278, that interact with the taccalonolides at sites relatively remote from the binding site (Fig. 7a)...mediating the post-reaction stability of the taccalonolide-tubulin complex" (l. 370-371). Previous work and Fig. 7a itself show the side chains of these three residues lining the binding site (and even a putative hydrogen bond involving H229). How can they be said to be remote if they actually make up the binding site?

Typo:

l. 442: An unanticipated finding from the generation of the Flu-tacca probes in this study is the identification a new strategy -> An unanticipated finding from the generation of the Flu-tacca probes in this study is the identification of a new strategy

Congratulations to the authors and best wishes for continuing success in this interesting project.

Federico Gago

We thank the reviewers for consideration of our revised manuscript and the additional comments. We have addressed these remaining points below and in the revised manuscript and SI document.

Reviewer #1 (Remarks to the Author):

The authors largely addressed the issues raised. However, the new data in Supplementary Figure 5 seem rather poor and uncontrolled. The amount of time for analysis is not mentioned, there is no $t = 0$ time point, and there is no indication of what percentage conversion the mass spec peaks might represent, given potentially different ionization potentials.

We have revised Supplementary Figure 5 to include the time that the compound was added to cells prior to analysis (8 hr) and have also expanded the x-axis of the LC/MS trace to include $t=0$ for the HPLC trace. We have also included $t=0$ traces for 10 prior to addition to cells (panel a), the hydrolysis product generated in the laboratory as a standard (panel d) as well as cells alone before addition of 10 (panel e). Together, these data show that the peak for the hydrolysis product is not observed with 10 prior to its addition to cells or in cells alone but is only observed after 10 has been incubated with cells.

One of the reasons that we have included two biological replicates in panels b and c is to demonstrate that, although the hydrolysis product predicted for 10 can indeed be extracted from cells after treatment with 10, there are differences between the ratios of these two compounds that are recovered even among biological replicates. Therefore, we are presenting these data as a qualitative assessment of the ability of 10 to be converted to the predicted hydrolysis product in the cells as opposed to a quantitative measurement of the rate or extent of conversion.

Reviewer #3 (Remarks to the Author):

Overall, the authors made a great effort to address all the issues raised by the reviewers and the revised version has improved its clarity and impact. However, I would like to add, for the sake of accuracy, a few comments/clarifications that may lead to very minor additional changes.

The first one refers to A2 in their response to reviewer #3's Q2. We are not talking here about the nominal resolution of an X-ray crystal structure versus that of a cryo-EM structure. What the reviewer was referring to was that in the 5EZY structure (which indeed "provides atomic resolution at 2.05 Å and allows unambiguous determination of the orientation and conformation of the ligand and binding pocket residues") some of the loops in beta-tubulin making up the binding site (and the M-loop in particular) have a conformation that is (necessarily) different from that known to exist in microtubules (cf. Fig. 2 [<https://link.springer.com/article/10.1007%2Fs10822-019-00208-w>] and Fig. S1 in ref. 26 [https://static-content.springer.com/esm/art%3A10.1007%2Fs10822-019-00208_w/MediaObjects/10822_2019_208_MOESM7_ESM.docx]). For this reason, since the fluorescent moiety is (very reasonably) proposed to interact with R278 and Q282 in the M-loop (Fig. 5e-h), the same docking experiment could have been done with PDB entries 3J6G ("optimized" by using restrained molecular dynamics simulations in D. Baker's group) or 5SYG (better nominal resolution but presenting gross steric clashes) [DOI: 10.1016/j.jmb.2017.01.001]. No further action is required.

We thank the reviewer for their comments and appreciate their expertise in the modeling field to help us more clearly represent this aspect of our work.

The second one is related to Q6 and A6 in their answer to the same reviewer. The terms "refine" and "refinement" are widely used by X-ray crystallographers in a very precise context; molecular modelers also "refine" a structure when they remove steric clash and optimize interactions during an energy minimization. What was done here was to "prepare" the coordinates deposited with the PDB to carry out some molecular modeling tasks. For this reason I believed -and still believe- that the use of "refine" is somewhat inappropriate and misleading and therefore should be avoided. The text now reads "optimized" (l. 468) and this is entirely acceptable and later on "To further refine the structure, ..." (l. 473); in the present context I do not see an objection.

We have ensured that the terms refine and refinement have been replaced by optimize or optimization when referring to the modeling

Lastly, and in relation to Q8/A8, I still do not understand the sentence "...L217, H229, and R278, that interact with the taccalonolides at sites relatively remote from the binding site (Fig. 7a)...mediating the post-reaction stability of the taccalonolide-tubulin complex" (l. 370-371). Previous work and Fig. 7a itself show the side chains of these three residues lining the binding site (and even a putative hydrogen bond involving H229). How can they be said to be remote if they actually make up the binding site?

We have revised this statement to indicate the relationship of these residues more specifically to the site of covalent interaction as opposed to the 'binding site'.

Typo:

l. 442: An unanticipated finding from the generation of the Flu-tacca probes in this study is the identification a new strategy -> An unanticipated finding from the generation of the Flu-tacca probes in this study is the identification of a new strategy

We have corrected this typo

REVIEWERS' COMMENTS:

Reviewer #1 (Remarks to the Author):

Experimentally, it is not difficult to calibrate LC/MS and to work out extraction efficiency for quantitative information regarding drug metabolism. I won't belabor the point regarding Supp. Figure 5 since it's small compared to the work as a whole. I appreciate the inclusion of data from more than one experiment, which make a strong case for knowing percent conversion. The authors should get such experiments done for any follow-on papers on taccalonolides.

We appreciate the reviewer's comment and editorial requests and have addressed each in point-by-point responses below:

Point-by-point responses to reviewer's comments:

Reviewer #1 (Remarks to the Author):

Experimentally, it is not difficult to calibrate LC/MS and to work out extraction efficiency for quantitative information regarding drug metabolism. I won't belabor the point regarding Supp. Figure 5 since it's small compared to the work as a whole. I appreciate the inclusion of data from more than one experiment, which make a strong case for knowing percent conversion. The authors should get such experiments done for any follow-on papers on taccalonolides.

We appreciate the comments by the reviewer and agree that the LC/MS quantification would be theoretically applicable for this experiment. However, our observation of differences in relative levels of the intact and cleaved intracellular compounds among the two experiments is consistent with additional observations that cellular import and export mechanisms are also in play to confound these results. The cellular transport of the taccalonolides and the probes is a valuable endeavor that we are actively exploring in current and future studies. In light of the scope of this manuscript, we believe it is more appropriate to report only the qualitative data, which we are confident in and are also supported by several lines of evidence. We will continue to understand the mechanisms underlying the cellular transportation of taccalonolides both qualitatively and quantitatively in future studies as suggested by the reviewer.